# CROSS-VIEW OPEN-VOCABULARY OBJECT DETECTION IN AERIAL IMAGERY

## ABSTRACT

Traditional object detection models are typically trained on a fixed set of classes, limiting their flexibility and making it costly to incorporate new categories. Open-vocabulary object detection addresses this limitation by enabling models to identify unseen classes without explicit training. Leveraging pretrained models contrastively trained on abundantly available ground-view image-text classification pairs provides a strong foundation for open-vocabulary object detection in aerial imagery. Domain shifts, viewpoint variations, and extreme scale differences make direct knowledge transfer across domains ineffective, requiring specialized adaptation strategies. In this paper, we propose a novel framework for adapting open-vocabulary representations from ground-view images to solve object detection in aerial imagery through structured domain alignment. The method introduces contrastive image-to-image alignment to enhance the similarity between aerial and ground-view embeddings and employs multi-instance vocabulary associations to align aerial images with text embeddings. Extensive experiments on the xView, DOTAv2, VisDrone, DIOR, and HRRSD datasets are used to validate our approach. Our open-vocabulary model achieves improvements of +6.32 mAP on DOTAv2, +4.16 mAP on VisDrone (Images), and +3.46 mAP on HRRSD in the zero-shot setting when compared to finetuned closed-vocabulary dataset-specific model performance, thus paving the way for more flexible and scalable object detection systems in aerial applications.

## 1 INTRODUCTION

Over the last decade, object detection has seen remarkable progress, with models attaining high accuracy on predefined categories in closed-set settings He et al. (2017); Ren et al. (2015); Carion et al. (2020). However, real-world applications demand open-vocabulary object detection, where models must recognize and localize unseen object categories. The emergence of large-scale pretrained Vision-Language Models (VLMs) Radford et al. (2021); Jia et al. (2021) has significantly advanced open-vocabulary object detection, enabling models to detect novel categories beyond their training set. Among these, contrastive vision-language models Ranasinghe et al. (2023); Khan & Fu (2023); Ak et al. (2024); Cui et al. (2022) such as OWLv2 Minderer et al. (2023) leverage large-scale image-text pairings to learn generalizable representations, facilitating zero-shot object detection and retrieval. While these models achieve strong performance in ground-view imagery, extending them to aerial-view object detection remains a fundamental challenge due to domain shifts and viewpoint variations. Aerial imagery introduces unique complexities: extreme scale variations, occlusions, and drastic perspective distortions, making direct transfer of ground-view models ineffective.

Researchers have developed ground-to-aerial domain adaptation techniques to leverage the abundance of labeled ground-level image data for aerial object detection. Existing approaches address the domain gap through generative modeling Zeng et al. (2024); Ma et al. (2024); Soto et al. (2020); Ye et al. (2024); Lu et al. (2020), adversarial training Chen et al. (2021); Wozniak et al. (2024), self-supervised learning Scheibenreif et al. (2024); Wang et al. (2024a), viewpoint-dependent feature matching Mule et al. (2025); Hou et al. (2023); Shan et al. (2014); Shugaev et al. (2024); Regmi & Shah (2019), and knowledge distillation Yao et al. (2024); Wang et al. (2024b).

Generative models, such as GANs, synthesize aerial-like images from ground-view data, expanding training datasets and enhancing model generalization. Adversarial domain adaptation reduces domain shift by training a discriminator to distinguish between ground and aerial feature distributions while guiding the model to learn domain-invariant representations. Self-supervised learning lever-

ages the structure of unlabeled aerial images through pretext tasks like rotation prediction, enabling the model to learn useful representations without explicit supervision. Viewpoint-dependent feature matching establishes correspondences between ground and aerial imagery by leveraging geometric relationships, ensuring better feature alignment. Lastly, knowledge distillation transfers information from a ground-view teacher model to a student model optimized for aerial detection, facilitating effective cross-domain knowledge transfer.

Beyond domain adaptation, research in open-vocabulary aerial detection aims to recognize objects outside the set of categories encountered during training. CastDet Li et al. (2024) utilizes a self-learning framework based on a student-teacher paradigm, integrating RemoteCLIP Liu et al. (2024a) to generate class-agnostic region proposals and pseudo-labels, thereby enabling open-vocabulary detection in aerial imagery. Another approach, SS-OWFormer Mullappilly et al. (2024), designed for satellite imagery, employs a feature-alignment mechanism alongside pseudo-labeling to enhance the detection of previously unseen objects. However, these methods heavily depend on large-scale aerial datasets for finetuning, pseudo-labeling, or feature alignment, making them reliant on labeled aerial supervision. In practice, such large-scale labeled datasets are often scarce or unavailable in the aerial domain, which further limits the applicability of these approaches.

In contrast, we propose a novel approach leveraging the abundance of ground-view image-text data available and align it with aerial representations using a contrastive learning framework, as seen in Fig. 1. Unlike approaches that require dataset-specific finetuning, which can lead to catastrophic forgetting of ground-view knowledge (refer Fig. 2), our contrastive learning formulation preserves pretrained vision-language model knowledge while ensuring robust cross-view feature alignment. Our vocabulary expansion enhances zero-shot detection of novel objects, making it more scalable and adaptable for open-vocabulary aerial object detection compared to existing approaches.

The key contributions of this work are as follows: **a. Contrastive Image-to-Image Alignment:** We introduce a novel contrastive alignment strategy that enhances semantic consistency between aerial-view and ground-view embeddings, facilitating the transfer of meaningful features from ground-view pretrained vision-language models (VLMs) to open-vocabulary object detection in aerial imagery. **b. Multi-Instance Vocabulary Association:** We propose a multi-instance vocabulary association technique that aligns aerial images with text embeddings, effectively leveraging textual descriptions to improve recognition of unseen categories in aerial view images. **c. Extensive Benchmark Evaluation:** We validate experiments on the xView Lam et al. (2018), DOTAv2 Ding et al. (2021), VisDrone Cao et al. (2021), DIOR Li et al. (2020), and HRRSD Zhang et al. (2019) datasets, achieving significant improvements in open-vocabulary detection performance, highlighting the scalability and effectiveness of our method.

Figure 1: **Motivation:** Ground-view contrastive pretrained open vocabulary detectors (top-left) fail to generalize to aerial views, and traditional finetuning (top-right) results in misaligned feature spaces. Our method (bottom-left) enforces cross-view contrastive alignment and aerial view-text associations, ensuring better semantic consistency and generalization for open-vocabulary aerial object detection (bottom-right). _Note_: Different object classes are color-coded: Airplane, Car, Truck. Different domains are shape-coded: ▲ Ground-view, ■ Text, ★ Aerial-view.

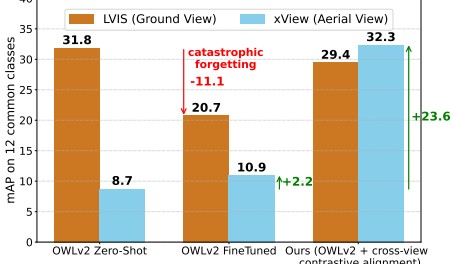

Figure 2: Our approach improves aerial-view detection while preserving ground-view performance and avoids catastrophic forgetting typical of naive finetuning.

## 2 RELATED WORK

**Aerial Object Detection** Deep learning has significantly advanced object detection in natural/ground-view images, leading to the adaptation of frameworks like Faster R-CNN Girshick (2015), RetinaNet Lin et al. (2017), YOLO Redmon et al. (2016), and DETR Zhu et al. (2020) for aerial imagery. However, aerial object detection faces challenges such as extreme scale variations, arbitrary orientations, and high object density. To address these challenges, methods like ROI-Transformer, R3Det Yang et al. (2021), and RSDet++ Qian et al. (2022) improve localization accuracy by using oriented bounding boxes, while techniques such as SCRDet Yang et al. (2019b) and ClusDet Yang et al. (2019a) enhance the detection of small and densely packed objects. Frameworks like Focus-and-Detect Koyun et al. (2022) and PARE-YOLO Zhang et al. (2025) further refine small object detection through clustering and multi-scale feature fusion. NavBLIP Li et al. (2025) integrates visual and contextual information for UAV-based detection, enhancing performance in dynamic environments. Despite these advancements, most methods operate under a closed-set assumption, meaning they are trained and evaluated on a fixed set of object categories. Expanding such models to recognize novel categories requires collecting large-scale annotated datasets, a costly and time-intensive process.

**Open-Vocabulary Object Detection** (OVD) Early work such as OVR-CNN Zareian et al. (2021) used bounding box annotations for a limited set of objects while leveraging image-caption pairs to extend the detection vocabulary. With the advent of pretrained Vision-Language Models (VLMs) like CLIP Radford et al. (2021) and ALIGN Jia et al. (2021), recent OVD approaches have transferred text-image knowledge into detection models using prompt learning and region-level finetuning. Methods like ViLD Gu et al. (2021) employ vision-language distillation to align textual and visual features, RegionCLIP Zhong et al. (2022) aligns region-level representations with textual concepts, and Detic Zhou et al. (2022) enhances detection capabilities by incorporating large-scale image classification datasets. Further refinements, including PromptDet Feng et al. (2022) and DetPro Du et al. (2022), optimize prompt embeddings for better visual-textual alignment. Additionally, models like YOLO-World Cheng et al. (2024) and F-VLM Kuo et al. (2022) address open-vocabulary detection challenges. YOLO-World integrates real-time vision-language modeling with Ultralytics YOLOv8, enabling efficient object detection based on descriptive text. F-VLM simplifies the training process by using frozen vision and language models, achieving state-of-the-art results with reduced computational overhead. Despite these advancements, applying OVD to aerial imagery remains challenging due to smaller dataset sizes and fundamental differences from ground-view images.

## 3 APPROACH

Recent advancements in vision-language models (VLMs) have significantly improved open-vocabulary object detection by aligning images and text in a shared embedding space. Our framework builds upon this idea to tackle cross-view open-vocabulary object detection, specifically aligning aerial-view images with ground-view and textual representations. Our model consists of a vision encoder $f_\theta$ and a text encoder $h_\phi$, which respectively project aerial images and textual queries into a common feature space. Given an aerial image $I_A$, the vision encoder extracts its feature representation $f_\theta(I_A) \in \mathbb{R}^d$, where $d$ is the size of the embedding space. Similarly, a textual query $Y$ is tokenized and processed by the text encoder $h_\phi$, producing a textual feature embedding $h_\phi(Y) \in \mathbb{R}^d$.

To ensure effective alignment between aerial, ground, and textual representations, our model incorporates two key learning mechanisms: Cross-View Representation Alignment (Sec. 3.1) and Aerial-Text Multi-Instance Association (Sec. 3.2). The success of these alignment strategies depends on the quality of the generated contrastive alignment data. To construct this, we introduce a data generation pipeline comprising Aerial-Ground Object Detection Correspondence (Sec. 3.3) and Aerial-Text Vocabulary Expansion (Sec. 3.4).

### 3.1 CROSS-VIEW REPRESENTATION ALIGNMENT

Standard contrastive vision-language models struggle to generalize from ground-view pretraining to aerial views due to the modality gap introduced by viewpoint shifts and domain discrepancies. Instead of direct finetuning on aerial imagery, we propose a cross-view contrastive alignment strategy that explicitly bridges the feature spaces of aerial and ground representations.

Given a batch of aerial images $I_A$ and corresponding ground-view references $I_G$, we apply contrastive learning to enforce similarity between aerial-ground pairs, while simultaneously pushing

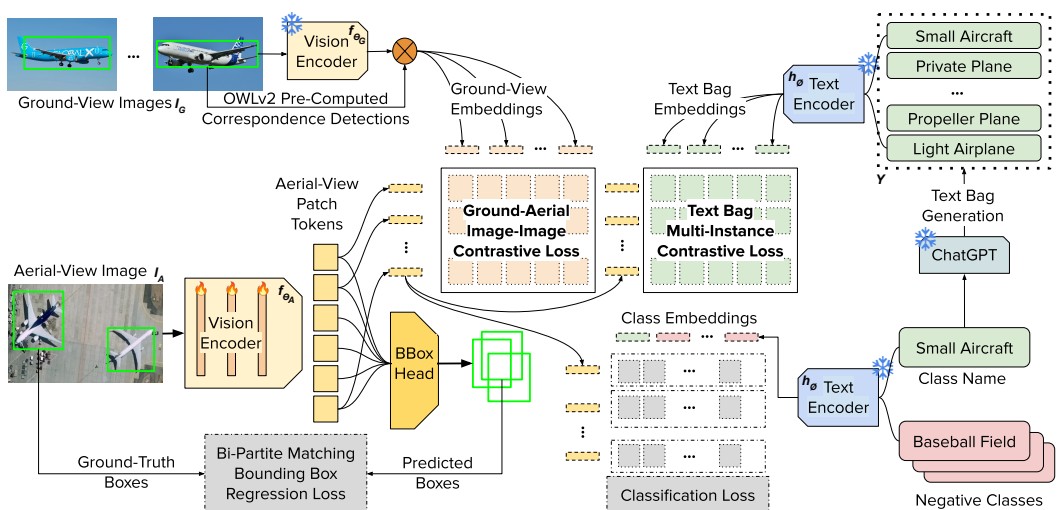

Figure 3: **Overview** Traditional open-vocabulary object detector finetuning includes two losses: a bi-partite matching based bounding box regression loss and a classification loss computed in the shared image–text embedding space. We further introduce two new components: a ground–aerial contrastive loss that aligns aerial and ground image embeddings, and a text-bag multi-instance contrastive loss that aligns aerial features with text bags. To realize these objectives, we generate an aerial–ground correspondence dataset providing cross-view positives/negatives, and also expand the class vocabulary by using ChatGPT to synthesize variants of class names ("text bags") consumed by the text encoder. Joint optimization over these losses places aerial ↔ ground ↔ text representations in a cohesive shared embedding space, yielding stronger zero-shot generalization in aerial imagery.

apart non-matching samples. Since only the aerial-view encoder is finetuned during training, we use a single notation $f_\theta$ to refer to the model being optimized, simplifying the formulation:

$$\mathcal{L}_{\textit{Image}_A\text{-}\textit{Image}_G} = -\frac{1}{N} \sum_{i=1}^{N} \log \frac{\exp(f_\theta(I_A^i)^\top, f_\theta(I_G^i))/\rho}{\sum_{j=1}^{N} \exp(f_\theta(I_A^i)^\top, f_\theta(I_G^j))/\rho}, \tag{1}$$

where $N$ represents the batch size, $i$ indexes a positive (correctly matched) aerial-ground pair, and $j$ indexes all samples in the batch. The numerator maximizes the similarity between the correct aerial-ground pair $(I_A^i, I_G^i)$, while the denominator contrasts it against all ground-view images $I_G^j$ in the batch, encouraging better separation between matching and non-matching samples. Here, $\rho$ is a temperature parameter controlling the sharpness of the similarity distribution.

By omitting a subscript for $f_\theta$, we emphasize that only one encoder is being finetuned (i.e., the aerial-view encoder), while the ground-view embeddings remain fixed, ensuring efficient adaptation of aerial representations to the pretrained feature space.

### 3.2 AERIAL-TEXT MULTI-INSTANCE ASSOCIATION

To further improve generalization, we introduce multi-instance contrastive learning between aerial images and their corresponding textual descriptions. Rather than enforcing a strict one-to-one correspondence, we adopt a multi-instance text association approach, where each aerial image $I_A^i$ is linked to a set of semantically related textual queries via a text-bag representation. This formulation enables better alignment between aerial imagery and natural language descriptions by accounting for ambiguities and variations in object naming conventions.

Our objective function extends MIL-NCE Miech et al. (2020), optimizing the alignment between aerial images and a diverse set of textual candidates:

$$\mathcal{L}_{\textit{Image}_A\text{-}\textit{Text}} = -\frac{1}{N} \sum_{i=1}^{N} \log \frac{\sum_k \sum_n \exp(h_\phi(Y_{k,n})^\top f_\theta(I_A^i)/\sigma)}{\sum_j \sum_n \exp(h_\phi(Y_{j,n})^\top f_\theta(I_A^i)/\sigma)}, \tag{2}$$

where $N$ represents the batch size, index $i$ refers to an aerial image $I_A^i$, while $j$ iterates over all possible textual descriptions in the batch. The index $k$ corresponds to a positive text-bag group,

which contains multiple semantically related textual descriptions for $I_A^i$, and $n$ indexes the individual text descriptions within the text-bag. The numerator sums over multiple positive textual descriptions $Y_{k,n}$ associated with $I_A^i$, while the denominator contrasts it against all text samples $Y_{j,n}$ in the batch, ensuring that the model learns to distinguish between relevant and irrelevant textual associations. This multi-instance contrastive loss promotes robust open-vocabulary object detection.

## 3.3 AERIAL-GROUND OBJECT DETECTION CORRESPONDENCE

In cross-view object detection, establishing category-based correspondence between aerial and ground-view detections is crucial for model generalization. Our data generation pipeline constructs aerial-ground correspondences by leveraging existing datasets (namely xView Lam et al. (2018) for aerial and LVIS Gupta et al. (2019) / CC12M Changpinyo et al. (2021) for ground) and enhancing them with inferred-labeling techniques. The goal is to align aerial detections with semantically corresponding ground-view samples while ensuring sufficient diversity. To facilitate cross-view representation learning for open-vocabulary aerial object detection, we construct a dataset $D_{aligned}$ by establishing correspondences between aerial and ground-view images. This process consists of two primary

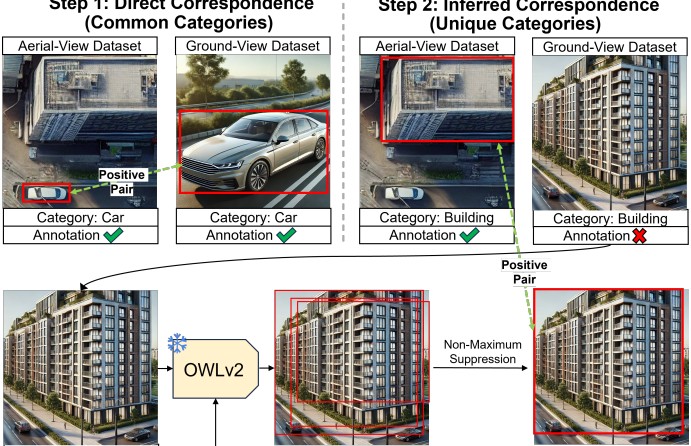

Figure 4: **Aerial-Ground Object Detection Correspondence** Step 1 (top-left) shows a case where the category 'Car' is annotated in both aerial and ground-view datasets, allowing direct positive pair generation using ground-truth annotations. Step 2 (top-right) illustrates a case where the category 'Building' lacks ground-view annotations. To address this, OWLv2 is used to generate detections in the ground-view image (bottom row), followed by non-maximum suppression to establish cross-view correspondence data for our contrastive alignment training.

steps: **Direct Correspondence**, as shown in Fig. 4, where detection matches between aerial and ground images are identified based on category, and **Inferred Correspondence**, where pseudo-matches are generated for categories that lack ground-view annotations. The final step involves **Augmentation and Normalization** to enhance the generated data's diversity and consistency.

**Direct Correspondence (For Common Categories)**: For object categories present in both the aerial and ground datasets, we directly extract aligned object instances. As seen in Algo. 1, for each category $c$ in the set of common categories $C_{common}$, we retrieve bounding box annotations $B_G$ and corresponding images $I_G$ from the ground-view dataset. These ground-view instances are then paired with their aerial counterparts, forming structured dataset entries $(B_A, B_G, y_c)$, where $B_A$ represents the aerial bounding box, $B_G$ is the corresponding ground bounding box, and $y_c$ is the class label. This process ensures precise alignment between aerial and ground perspectives, establishing a strong foundation for shared representations.

---

**Algorithm 1: Aerial-Ground Object Detection Correspondence**

1: **Initialize:** $D_{aligned} \leftarrow \emptyset$      # Empty dataset
  ▷ **Step 1: Direct Correspondence (Common Categories)**
2: **for** $c \in C_{common}$ **do**
3:      Extract $\{B_G, I_G\}$      # Ground Boxes, Images
4:      $D_{aligned} \leftarrow D_{aligned} \cup \{(B_A, B_G, y_c)\}$
5: **end for**
  ▷ **Step 2: Inferred Correspondence (Unique Categories)**
6: **for** $c \in C_{unique}$ **do**
7:      $\hat{B}_G \leftarrow$ OWLv2$(I_G)$      # Generate ground boxes
8:      $\hat{B}_G \leftarrow$ NMS$(\hat{B}_G, \tau)$      # Remove redundant
9:      $D_{aligned} \leftarrow D_{aligned} \cup \{(B_A, \hat{B}_G, y_c)\}$
10: **end for**
  ▷ **Step 3: Augmentation & Normalization**
11: $D_{aligned} \leftarrow$ Augment$(D_{aligned})$      # Crop, Rotate
12: $D_{aligned} \leftarrow$ Norm$(D_{aligned})$      # Feature Align
13: **Return** $D_{aligned}$      # Store for training

**Inferred Correspondence (For Unique Categories)**: For categories that exist in the aerial dataset but lack explicit ground-view annotations, we generate inferred correspondences using vision-language models. For each category $c \in C_{unique}$, we apply OWLv2 to detect objects in the ground-view dataset, producing an initial set of bounding boxes $\hat{B}_G$. To refine these detections, we employ Non-Maximum Suppression (NMS) with a confidence threshold $\tau$ to eliminate redundant or low-confidence detections, ensuring only reliable bounding boxes are retained. The final pseudo-ground view detections $\hat{B}_G$ are then associated with aerial objects, forming dataset pairs $(B_A, \hat{B}_G, y_c)$. This inferred alignment enables training on categories that lack explicit ground-truth annotations.

**Augmentation and Normalization**: To enhance dataset robustness and improve alignment consistency, we apply a series of augmentation and normalization techniques. Geometric transformations, such as cropping and rotation, diversify the dataset and mitigate overfitting. Additionally, feature normalization is applied to harmonize aerial and ground embedding spaces, ensuring better compatibility during model training. After these refinements, the final dataset $D_{\text{aligned}}$ is stored for training.

### 3.4 AERIAL-TEXT VOCABULARY EXPANSION

Here, we focus on enhancing category representation by expanding textual variations of aerial object categories. The primary goal is to improve the model's ability to recognize multiple textual references for the same object, enabling effective aerial object detection in open-vocabulary scenarios. The process begins by initializing an empty dataset $D_{\text{text}}$ to store textual variations of aerial object categories. As seen in Algo. 2, for each category $c$ in the aerial dataset $C_A$, ChatGPT is utilized to generate a set of label variations $\mathcal{V}_c$.

These variations include synonymous terms, alternative phrasings, and domain-specific terminology that may be used to refer to the same object in different contexts. The generated variations are stored in $D_{\text{text}}$ as pairs of the original category $c$ and its corresponding set of variations $\mathcal{V}_c$. Once all categories are processed, the dataset $D_{\text{text}}$ is finalized and returned for use in training. This expansion follows a hierarchical association where each category serves as a parent node with multiple semantically related variations acting as child nodes. By structuring the data in this way, the model can better interpret and generalize aerial object classes in an open-vocabulary setting.

It is worth noting that while our data generation pipeline (Aerial-Ground Object Detection Correspondence and Aerial-Text Vocabulary Expansion) provides essential correspondences and variations, the key novelty lies in how these resources are exploited through structured contrastive alignment, rather than in dataset construction itself.

---

**Algorithm 2: Vocabulary Expansion**

1: **Initialize:** $D_{\text{text}} \leftarrow \emptyset$     # Empty dataset
2: **for** $c \in C_A$ **do**
3:     $\mathcal{V}_c \leftarrow \text{ChatGPT}(c)$  # Generate label variations
4:     $D_{\text{text}} \leftarrow D_{\text{text}} \cup \{(c, \mathcal{V}_c)\}$
5: **end for**
6: **Return** $D_{\text{text}}$         # Store for training

---

| Category | Variations |
|---|---|
| Small Aircraft | Light airplane, Private plane, Single-engine aircraft, Propeller plane, Cessna-type aircraft, General aviation aircraft |
| Helicopter | Chopper, Rotary-wing aircraft, Rotorcraft, Helo, Air ambulance, Military helicopter, Rescue helicopter |
| Shed | Storage shed, Outbuilding, Toolshed, Garden shed, Small barn |
| Excavator | Digger, Backhoe, Trackhoe, Mechanical shovel, Hydraulic excavator |
| Small Car | Compact car, Hatchback, Subcompact vehicle, Economy car, City car, Two-door car, Four-door car |

Table 1: Vocabulary expansion examples.

## 4 EXPERIMENTS

We evaluate zero-shot transfer on unseen aerial benchmarks and compare our approach with state-of-the-art open-vocabulary detectors. We also carry out ablation studies to evaluate the impact of various design decisions, such as encoder patch size, loss contributions etc.

**Training Data.** Our pipeline builds cross-view correspondences and expands the vocabulary to strengthen contrastive alignment. Between xView and LVIS, 12 categories overlap and 48 are unique to xView. We create 25,403 positive aerial $\leftrightarrow$ ground pairs via direct matches, rising to 50,650 with inferred correspondences; adding CC12M brings the total to 310,548 pairs. Vocabulary expansion grows 60 aerial categories into 360 variants, broadening textual coverage (examples in Tab. 1).

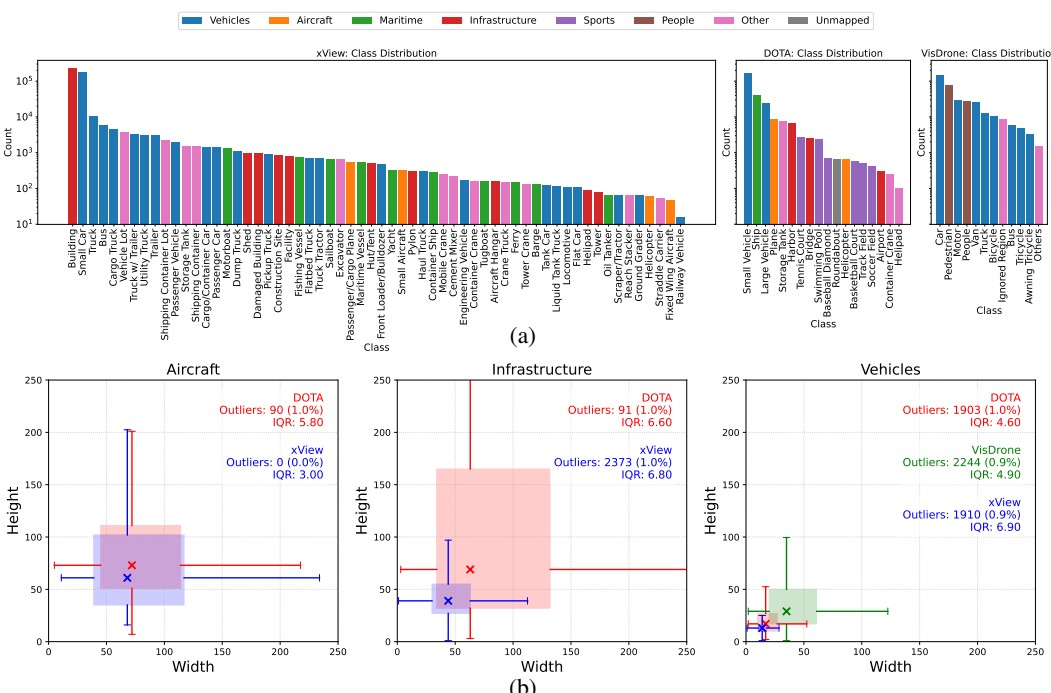

(a)

(b)

Figure 5: **Distribution** of **(a)** object categories and **(b)** size variations across three remote sensing datasets: xView, DOTAv2, and VisDrone (Images). The figure highlights category imbalance and significant object scale variations, reflecting challenges in open-vocabulary aerial object detection.

**Evaluation Datasets.** We evaluate generalization on five remote-sensing benchmarks: xView Lam et al. (2018), DOTAv2 Ding et al. (2021), VisDrone (images/videos) Cao et al. (2021), DIOR Li et al. (2020), and HRRSD Zhang et al. (2019). These datasets span satellite, aerial, and drone imagery. Collectively, they contain millions of labeled objects across dozens of categories with wide variation in scale, orientation, and scene complexity, stressing robustness to different sensors, viewpoints, conditions, and object sizes. Fig. 5 summarizes instance and size distributions, revealing pronounced category imbalance and scale variation, which are key challenges for open-vocabulary aerial object detection.

**Metrics** For our evaluation, we utilize the standard $AP_{50:95}$ metric, adhering to the benchmarking protocol set by MS COCO Lin et al. (2014). Further to assess overall performance across both seen and unseen object classes we utilize Harmonic Mean (HM) of $mAP_{base}$ and $mAP_{novel}$ classes.

### 4.1 Zero-Shot Generalization in Aerial Datasets

Our model is trained only on our generated xView–LVIS cross-view vocabulary-expanded data and evaluated in a zero-shot manner on DOTAv2, VisDrone, DIOR, and HRRSD, with *no* training data from those test datasets. In all experiments, we consistently refer to this protocol as *zero-shot transferability*. In Tab. 2, we compare our zero-shot object detection approach against YOLOv11 (finetuned) Jocher & Qiu (2024), which is trained on the entire dataset, across six aerial datasets: xView, DOTAv2, VisDrone (Images), VisDrone

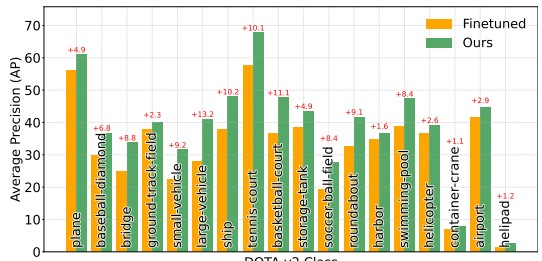

Figure 6: Per-class zero-shot performance of our model on DOTAv2 outperforms finetuning.

(Videos), DIOR, and HRRSD. In contrast, our model never sees these datasets during training, yet achieves consistent improvements in a zero-shot setting, with performance gains of **+6.32** on DOTAv2, **+4.16** on VisDrone (Images), **+2.37** on VisDrone (Videos), **+2.23** on DIOR, and **+3.46**

on HRRSD over YOLO. For xView, which is not evaluated in a true zero-shot setting (*), our approach still improves by **+2.99**. These results highlight our model's strong cross-domain generalization, demonstrating competitive performance without dataset-specific finetuning. Refer to Fig. 7 for qualitative results across datasets and Fig. 6 for the per-class breakdown on DOTAv2.

| Datasets | Finetuned | | Zero-Shot |
|---|---|---|---|
| | OWLv2 | YOLOv11 | Ours |
| xView Lam et al. (2018) | 12.65 | 34.92 | 37.91* (+2.99) |
| DOTAv2 Ding et al. (2021) | 15.01 | 32.28 | 38.60 (+6.32) |
| VisDrone (Images) Cao et al. (2021) | 25.70 | 40.81 | 44.97 (+4.16) |
| VisDrone (Videos) Cao et al. (2021) | 18.44 | 34.65 | 37.02 (+2.37) |
| DIOR Li et al. (2020) | 45.91 | 61.68 | 63.91 (+2.23) |
| HRRSD Zhang et al. (2019) | 53.25 | 70.66 | 74.12 (+3.46) |

Table 2: Comparison of OWLv2 (Finetuned), YOLO (Finetuned) and our zero-shot approach across multiple datasets. Note: '*' xView is not evaluated in a zero-shot setting.

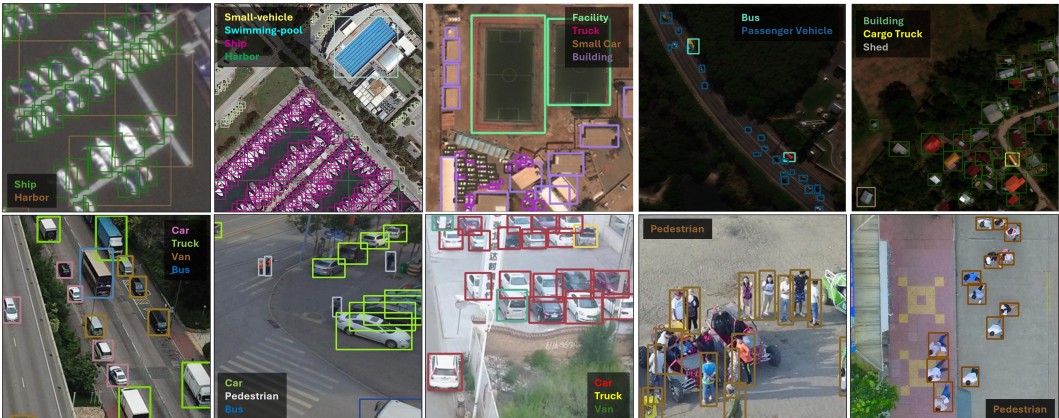

Figure 7: Qualitative results across diverse aerial dataset scenarios, including maritime environments (*1, 2*), sports facilities (*3*), nighttime conditions (*4, 5*), urban scenes featuring multiple interacting object classes (*6, 7, 8*), and pedestrians across widely varying viewpoints (*9, 10*).

| Method | External | $\mathcal{T}_{novel}$ | $\mathcal{T}_{cls}$ | mAP | mAP$_{base}$ | mAP$_{novel}$ | HM |
|---|---|---|---|---|---|---|---|
| Detic | NWPU-RESISC45 | ✗ | ✓ | 16.8 | 19.8 | 4.8 | 7.7 |
| ViLD | DIOR | ✓ | ✗ | 25.6 | 28.5 | 14.2 | 19.0 |
| OV-DETR | DIOR | ✓ | ✗ | 25.6 | 25.6 | 25.5 | 25.6 |
| BARON | NWPU-RESISC45 | ✗ | ✓ | 27.4 | 29.4 | 19.5 | 23.5 |
| YOLO-World | NWPU-RESISC45 | ✗ | ✓ | 32.9 | 39.1 | 8.5 | 13.9 |
| GroundingDINO | NWPU-RESISC45 | ✗ | ✓ | 33.0 | 40.5 | 3.3 | 6.1 |
| GLIP | NWPU-RESISC45 | ✗ | ✓ | 33.8 | 41.0 | 5.4 | 9.5 |
| CastDet | DIOR | ✗ | ✗ | 38.5 | 36.5 | 46.5 | 40.9 |
| CastDet | DIOR | ✓ | ✗ | 40.5 | 39.0 | 46.3 | 42.3 |
| *Ours* | xView | ✗ | ✗ | **44.9** | **42.7** | **49.2** | **45.7** |

Table 3: Performance comparison of SOTA open-vocabulary object detection methods on the VisDrone (Images) dataset. Here, $\mathcal{T}_{novel}$ indicates whether novel classes are pre-known, while $\mathcal{T}_{cls}$ denotes whether additional classification or caption datasets are used during training. The best method is shown in Red, and the second-best method is shown in Blue.

### 4.2 COMPARISON TO STATE-OF-THE-ART

As seen in Tab. 3, our method surpasses existing state-of-the-art open-vocabulary object detection approaches on the VisDrone (Images) dataset. These gains underscore the effectiveness of our aerial-ground alignment and aerial-text association.

### 4.3 ABLATION STUDIES

#### 4.3.1 VISION ENCODER PATCH SIZE

The ablation study in Tab. 4 assesses the impact of vision encoder patch size (Patch-16 vs. Patch-14) on open-vocabulary aerial object detection. Smaller patch sizes (Patch-14) consistently improve performance across all datasets, with DOTAv2 showing the largest gain of **+4.63**. This improvement is likely due to Patch-14 capturing finer spatial details, leading to better feature granularity and enhanced object discrimination, especially in high-variance aerial imagery.

| Datasets | Patch Size | |
|---|---|---|
| | *Ours (/16)* | *Ours (/14)* |
| xView | 36.48* | 37.91* (+1.43) |
| DOTAv2 | 33.97 | 38.60 (+4.63) |
| VisDrone (Images) | 42.66 | 44.97 (+2.31) |
| VisDrone (Videos) | 35.78 | 37.02 (+1.24) |

Table 4: Impact of encoder patch size on open-vocabulary aerial detection performance. Note: '*' indicates xView is not evaluated zero-shot.

#### 4.3.2 AERIAL-GROUND CORRESPONDENCE DATASET

Tab. 5 compares detection performance on xView and DOTAv2 when using different aerial-ground correspondence datasets (LVIS vs. CC12M) for alignment. CC12M leads to a substantial performance improvement across both datasets (**+3.23** on xView, **+6.87** on DOTAv2), highlighting the benefits of a larger and more diverse dataset for cross-view feature alignment. Aerial-ground alignment benefits from larger and more diverse datasets, as seen with CC12M outperforming LVIS. This suggests that training with a broader distribution of images improves cross-view generalization and enhances feature

| Evaluation Dataset | Aerial-Ground Correspondences | |
|---|---|---|
| | *LVIS* | *CC12M* |
| xView | 33.25 | 36.48 |
| DOTAv2 | 27.10 | 33.97 |

Table 5: Performance comparison on xView and DOTAv2 using different datasets (LVIS, CC12M) as the source of aerial-ground correspondence.

transferability, particularly for open-vocabulary aerial detection. The effect is more pronounced on DOTAv2, due to its higher scene complexity, where additional aerial-ground associations provide stronger supervision.

#### 4.3.3 CONTRIBUTION OF CONTRASTIVE LOSSES

The individual contributions of $Image_A$-$Text$ and $Image_A$-$Image_G$ contrastive losses to overall cumulative performance on the xView and DOTAv2 datasets are evaluated in Tab. 6. The $Image_A$-$Text$ contrastive loss aligns aerial image features with textual representations, facilitating open-vocabulary recognition, whereas the $Image_A$-$Image_G$ contrastive loss

| Datasets | Contrastive Losses | | |
|---|---|---|---|
| | $Image_A$-$Text$ | $Image_A$-$Image_G$ | *Both* |
| xView | 26.13 | 32.41 | 33.25 |
| DOTAv2 | 19.98 | 25.06 | 27.10 |

Table 6: Impact of individual losses.

enforces aerial-to-ground view consistency, improving domain adaptation. The cumulative score reflects the combined effect of both losses, highlighting that $Image_A$-$Image_G$ contrastive learning plays a more dominant role in feature alignment, as indicated by its higher individual contribution. However, the synergy between both losses leads to the best overall performance, demonstrating the importance of multi-modal representation learning in cross-view object detection.

## 5 CONCLUSION

We propose a novel cross-view contrastive alignment framework for open-vocabulary object detection in aerial imagery. To facilitate this, we establish aerial-ground correspondences using xView, LVIS, and CC12M, generating a large corpus of correspondence pairs. To bridge the modality gap between aerial and ground views, we establish cross-view alignment by learning semantic correspondences between detected objects across perspectives. Furthermore, we enhance generalization by employing multi-instance contrastive learning to align aerial images with their textual descriptions. Through extensive experiments, we demonstrate that our approach in a zero-shot setting outperforms finetuned models, advancing the scalability and adaptability of aerial object detection.

**Ethics Statement**   This work uses only publicly available datasets (xView, DOTAv2, VisDrone, DIOR, HRRSD, LVIS, CC12M) and involves no human-subject studies or personally identifiable information. All third-party data has been used under their original licenses, with citations provided in the paper. Our models are trained and evaluated for research on open-vocabulary object detection in aerial imagery; they are *not* intended for real-time surveillance or downstream uses that could infringe on privacy or civil liberties.

**Reproducibility Statement**   We provide details on all components needed to reproduce our results. Model objectives and notation are specified in Sec. 3, including the image-image contrastive loss (Eq. 1) and the aerial-text multi-instance objective (Eq. 2). The data-construction pipeline is detailed in Secs. 3.3-3.4, with algorithmic steps for aerial-ground correspondence (Algo. 1) and vocabulary expansion (Algo. 2). Training data composition, evaluation datasets, zero-shot protocol, and metrics are described in Sec. 4. Upon acceptance, we will provide public access to code, trained checkpoints, and exact commands to regenerate all tables and figures, including: (i) the aerial-ground correspondence pipeline with thresholds, sampling ratios, and random seeds; (ii) vocabulary-expansion lists and prompt templates; (iii) full training recipes (frozen modules, optimizer, learning rate, weight decay, temperatures, batch sizes), evaluation prompts, and image resolutions; and (iv) per-experiment configuration files for zero-shot and fine-tuned baselines.

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

APPENDIX

## A    MODEL-AGNOSTIC CONTRASTIVE ALIGNMENT STRATEGY

Experimental results on the VisDrone dataset demonstrate the model-agnostic nature of our framework, as shown using OWLv2 and GroundingDINO. Given any open-vocabulary ground-view detector, we generate cross-view data which, combined with our contrastive alignment strategy, enables the construction of an open-vocabulary aerial-view detector—without being tied to a specific architecture like OWLv2. As seen in 7, the framework transfers seamlessly across models and facilitates effective ground-to-aerial adaptation.

| Model | Finetuned | Ours *mAP* |
|---|---|---|
| OWLv2 Minderer et al. (2023) | 25.70 | 44.97 (+19.27) |
| GroundingDINO Liu et al. (2024b) | 33.00 | 47.15 (+14.15) |

Table 7: Model-agnostic results on the VisDrone dataset.

## B    QUALITATIVE RESULTS

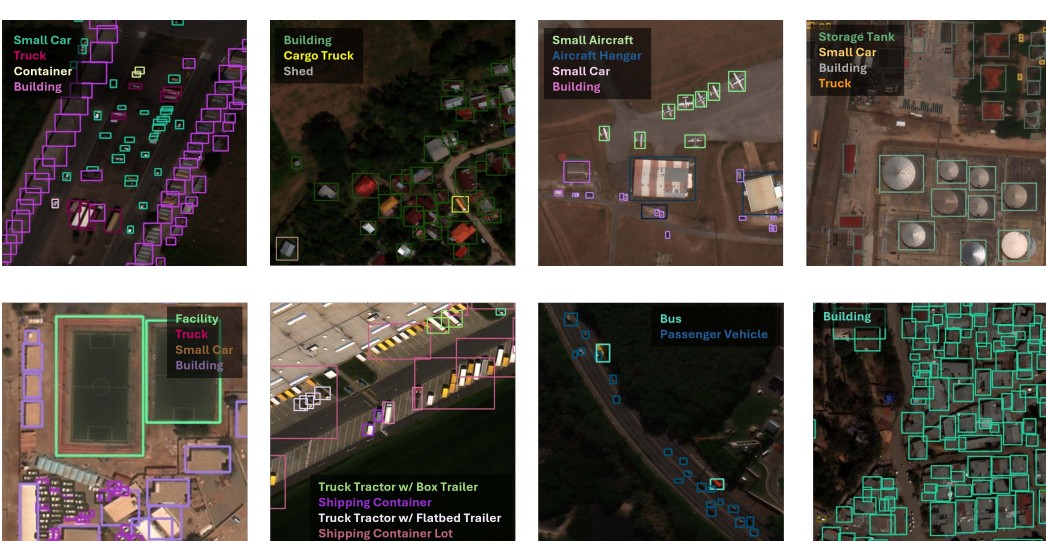

Figure 8: Qualitative results on xView Lam et al. (2018)

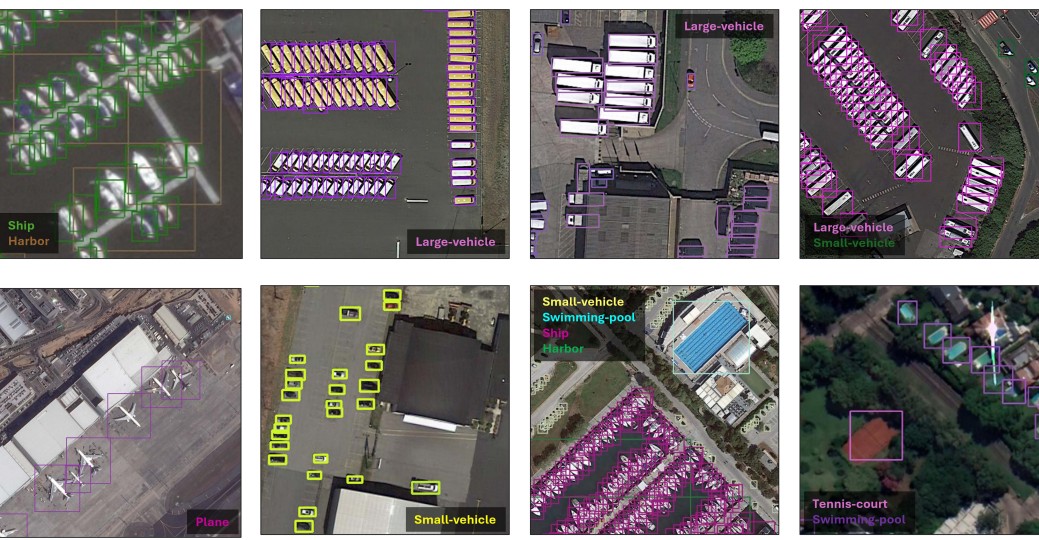

Figure 9: Qualitative results on DOTAv2 Ding et al. (2021)

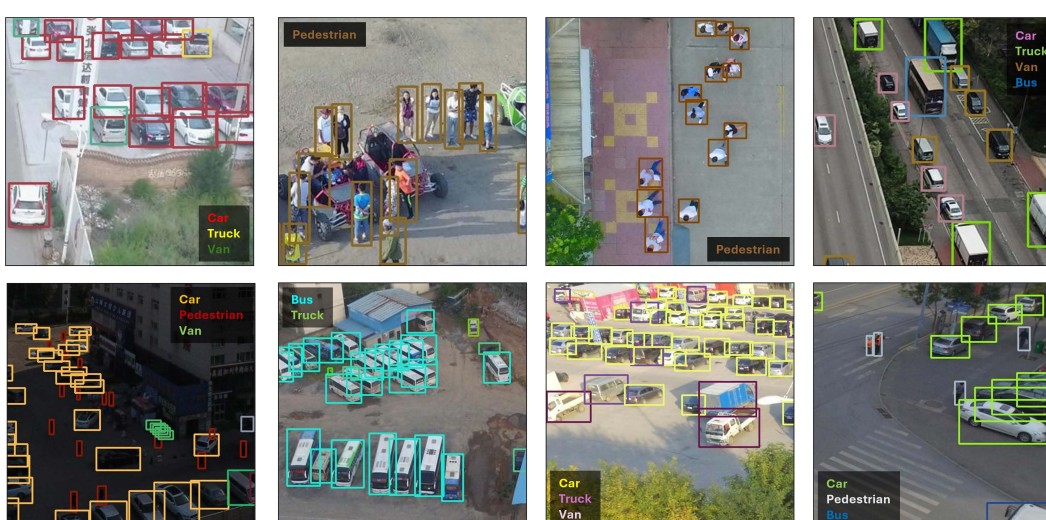

Figure 10: Qualitative results on VisDrone Cao et al. (2021)

## C  ANALYSIS OF PSUEDO-LABEL QUALITY

| Category | Precision | Category | Precision |
|---|---|---|---|
| Mobile Crane | 0.375 | Shipping Container | 0.400 |
| Passenger Car (Rail) | 0.077 | Facility | 0.625 |
| Oil Tanker | 0.364 | Truck w/Box | 0.875 |
| Storage Tank | 0.125 | Small Car | 0.800 |
| Excavator | 0.667 | Shed | 0.600 |
| Shipping Container Lot | 0.091 | Construction Site | 0.333 |
| Utility Truck | 0.333 | Sailboat | 0.833 |
| Aircraft Hangar | 0.273 | Cargo Car (Rail) | 0.444 |
| Container Crane | 0.769 | Haul Truck | 0.818 |
| Tank Car (Rail) | 0.714 | Motorboat | 0.875 |
| Fishing Vessel | 0.071 | Damaged Building | 0.800 |
| Truck w/Flatbed | 0.500 | Straddle Carrier | 0.091 |
| Tower Crane | 0.700 | Container Ship | 0.429 |
| Cement Mixer | 0.364 | Pickup Truck | 0.500 |
| Front Loader/Bulldozer | 0.600 | Tugboat | 0.444 |
| Helipad | 0.500 | Crane Truck | 0.667 |
| Dump Truck | 0.400 | Passenger Vehicle | 1.000 |
| Reach Stacker | 0.333 | Building | 1.000 |
| Cargo Truck | 0.750 | Barge | 0.714 |
| Scraper/Tractor | 0.500 | Truck Tractor | 0.714 |
| Pylon | 0.545 | Bus | 0.857 |
| Flat Car (Rail) | 0.182 | Truck w/Liquid | 0.400 |
| Vehicle Lot | 0.833 | Ferry | 0.833 |
| Cargo Plane | 0.444 | Hut/Tent | 0.889 |
| Small Aircraft | 0.900 | Ground Grader | 0.375 |
| Helicopter | 0.714 | Trailer | 0.909 |
| Tower | 0.700 | Locomotive | 0.625 |
| Railway Vehicle | 0.700 | Yacht | 0.714 |

Table 8: Psuedo-label precision by category (default OWLv2 detection confidence threshold = 0.3 ).

A class balanced sample of 500 psuedo-detections were annotated by hand to verify the correctness. Of these 500 annotations, 261 predictions (55.8%) were fully correct at the parent-category level. This was based on the default confidence threshold of 0.3. The precision can be further boosted to 70% by filtering the detections with a threshold, of 0.45, as can be seen in Figure 11. Classwise precisions (at default threshold = 0.3) are also also reported in Table 8.

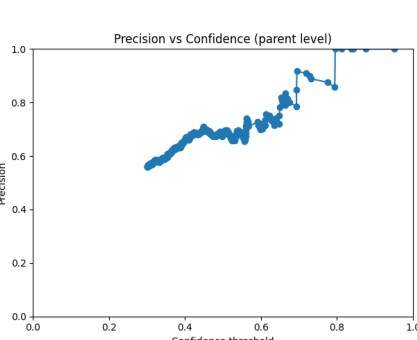

Figure 11: Precision vs Detection Confidence threshold from OWLv2.

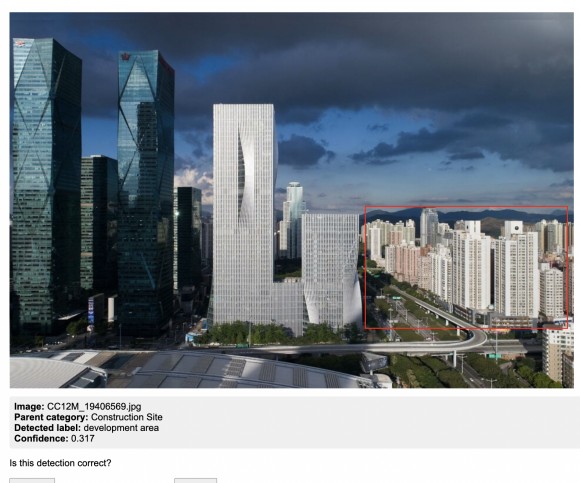

Figure 12: Screenshot of Annotation Tool

## D  CROSS-VIEW OPEN-VOCABULARY DETECTION FRAMEWORK OVERVIEW

As illustrated in 13, our cross-view open-vocabulary detection framework is organized into four stages that together connect aerial and ground-view images with text in a unified embedding space.

**(1) Aerial-Ground Object Detection Correspondence.**   We first construct a corpus of aligned aerial-ground pairs by combining xView aerial images with LVIS and CC12M ground-view imagery. For categories that are shared between xView and LVIS, we use ground-truth bounding boxes to form direct correspondences between aerial instances and ground-view crops. For categories that are present only in xView, we generate pseudo ground-view boxes by running an open-vocabulary detector (OWLv2) on LVIS/CC12M images, followed by non-maximum suppression and confidence filtering to remove duplicate or low-quality predictions. The resulting dataset contains both direct and inferred correspondences and is further augmented via geometric transformations (e.g., cropping, rotation) and feature normalization to improve robustness and balance across categories.

**(2) Aerial-Text Vocabulary Expansion.**   To better cover the long-tail variation in how aerial objects are described, we expand each xView category into a semantically rich "text bag." For every base class label (e.g., *small aircraft*, *shed*, *excavator*), we use ChatGPT to generate multiple paraphrases and synonyms, including generic terms, fine-grained variants, and domain-specific phrases. These variants are treated as a set of positive textual descriptions for the same underlying category. During training, the text encoder embeds all phrases in a given text bag, allowing the model to associate an aerial region with any of its plausible text concepts rather than a single canonical label.

**(3) Training with Cross-View and Cross-Modal Objectives.**   Given the correspondence dataset and expanded text vocabulary, we train the aerial encoder on three complementary losses. First, we retain the standard detection objective used by the underlying open-vocabulary detector, consisting of bipartite matching, bounding-box regression, and classification in the shared image-text embedding space. Second, we add an image-image contrastive loss that pulls together embeddings of matched aerial and ground-view crops while pushing apart non-matching pairs. This loss is applied only to the aerial encoder, which is finetuned to align with the frozen ground-view feature space. Third, we introduce an image-text multi-instance (MIL) contrastive loss: for each aerial crop, all phrases in its associated text bag are treated as potential positives, and the model maximizes aggregated similarity against a batch of negatives drawn from all other classes. Joint optimization over these objectives places aerial, ground, and text representations in a coherent shared space.

**(4) Zero-Shot Inference on Unseen Aerial Datasets.**   At test time, only the aerial encoder and text encoder are used. We provide class names (and optionally their expanded text bags) for the target

dataset as text prompts, and the aerial encoder processes each input image to produce region-level features. Detection proceeds by scoring each region against the text embeddings, followed by non-maximum suppression to obtain final boxes and labels. Crucially, no images or annotations from the evaluation datasets (DOTAv2, VisDrone, DIOR, HRRSD) are used during training; the model relies entirely on the cross-view correspondences and vocabulary expansion constructed from xView, LVIS, and CC12M. This design enables genuine zero-shot transfer to new aerial benchmarks while preserving the open-vocabulary capabilities of the underlying ground-view detector.

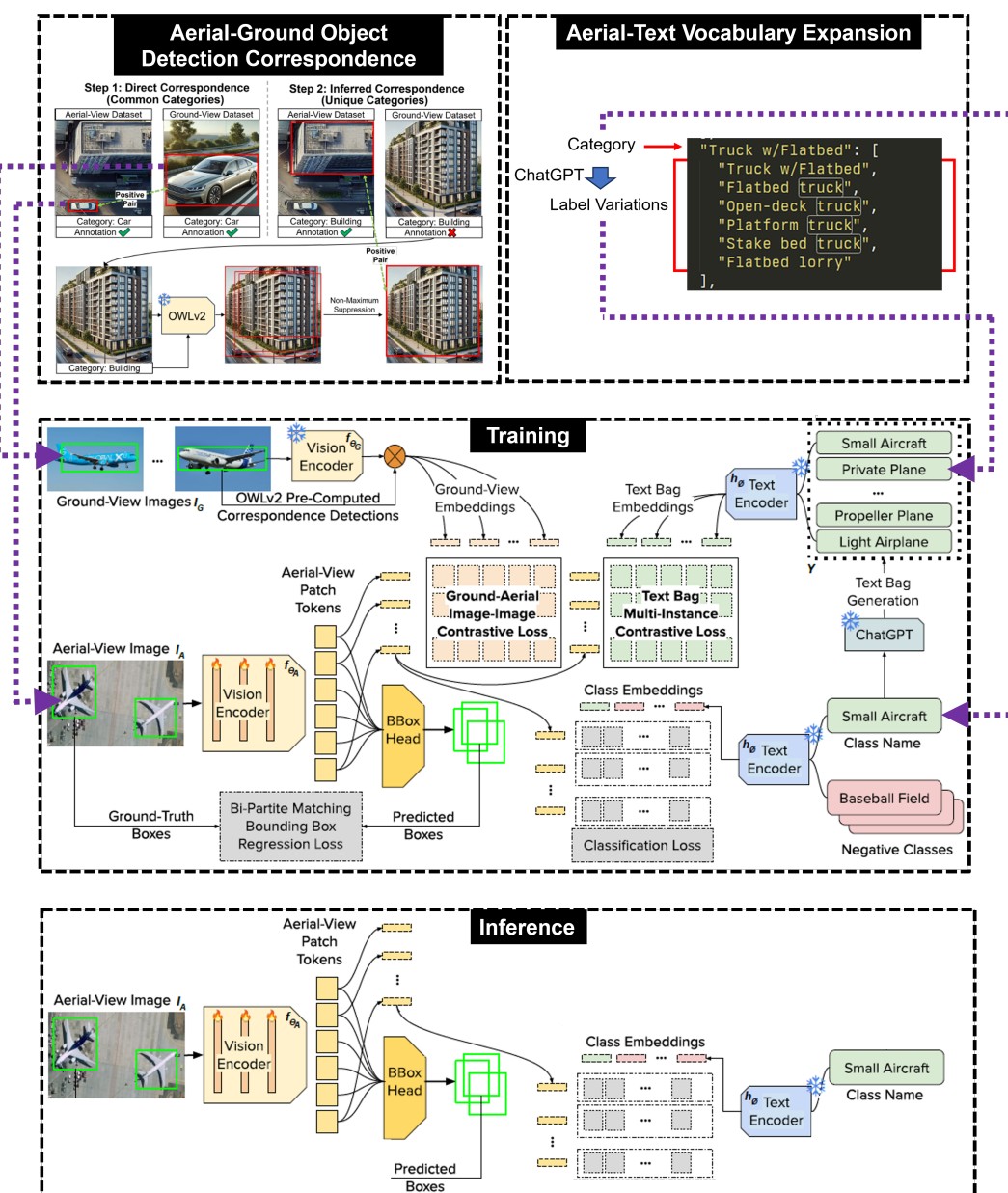

Figure 13: **Overall pipeline of our cross-view open-vocabulary object detection framework.** Training leverages aerial–ground correspondences and expanded text-bag to learn aligned image–image and image–text embeddings. During inference, the model uses only the aerial encoder and text prompts to perform zero-shot detection on unseen aerial datasets.

Table 9: Model architecture, training configuration, evaluation setup, and inference speed used in our experiments, unless otherwise specified.

| Component | Setting |
|---|---|
| Backbone | OWLv2 (ViT-L/14, ViT-B/16) |
| Training data size | 310,548 aerial–ground correspondence pairs |
| Image resolution | $960 \times 960$ |
| Batch size | 4 |
| Optimizer | AdamW |
| Learning rate | $1 \times 10^{-4}$ |
| Weight decay | 0.05 |
| Max training iterations | 60,000 |
| Image–image contrastive temperature ($\rho$) | 0.07 |
| Image–text contrastive temperature ($\sigma$) | 0.02 |
| Detection confidence threshold | 0.3 (OWLv2 default) |
| Zero-shot evaluation datasets | DOTAv2, VisDrone, DIOR, HRRSD |
| Inference speed (FPS) | 5.2 (ViT-L/14), 10.8 (ViT-B/16) on A100 |

## E  ADDITIONAL IMPLEMENTATION AND EVALUATION DETAILS

We have corrected citation formatting issues and ensure that all experimental tables explicitly state the evaluation units (e.g., mAP). For clarity and to facilitate fair comparison and future work, Table 9 summarizes our model architecture, training setup, evaluation benchmarks, and inference speed.

**Detection thresholds and correspondences.**  Unless otherwise stated, we apply the default OWLv2 detection confidence threshold of 0.3 when building correspondences between predictions and ground-truth annotations.

**Handling of VisDrone videos.**  For VisDrone, we evaluate each video by treating it as a sequence of independent frames: the model processes each frame individually, and we do not use any explicit temporal aggregation or tracking across frames.

**DOTAv2 evaluation protocol.**  On DOTAv2, we report results using horizontal bounding boxes (HBBs). Specifically, for each oriented bounding box (OBB) we construct the minimum axis-aligned rectangle that fully encloses the OBB, and compute all metrics on these derived HBBs.

## F  TEXT-BAG CONSTRUCTION AND SENSITIVITY ANALYSIS

**Multi-instance text-bag construction.**  For each category, we construct a multi-instance text-bag by augmenting the base class name with approximately 4–9 ChatGPT-generated synonyms. These synonyms are sampled once and fixed for all experiments in the main paper. The text-bag is then used as the pool of textual queries associated with that category during training and evaluation.

Table 10 provides a subset of the category–synonym mappings used in our experiments.

**Sensitivity to the number of synonyms per class.**  To assess the sensitivity of our method to the text-bag design, we vary the number of synonyms per category while keeping all other training settings fixed. Specifically, we compare:

- $k = 3$: a subsampled text-bag with three randomly selected synonyms per category;

- $k = $ full: the default 4–9 synonyms per category used in the main paper;

- $k = 9$: a capped setting where we use up to nine synonyms per category (padding with all available synonyms when fewer than nine are defined).

Table 10: Example category–synonym mappings used in the main paper. Each category is associated with a text-bag of ChatGPT-generated synonyms.

| Category | Synonyms |
|---|---|
| Small Aircraft | Light airplane, Private plane, Single-engine aircraft, Propeller plane, Cessna-type aircraft, General aviation aircraft, Small fixed-wing plane |
| Cargo Plane | Freight aircraft, Transport plane, Cargo jet, Cargo transporter, Military cargo plane, Heavy-lift aircraft, Air freighter |
| Helicopter | Chopper, Rotary-wing aircraft, Rotorcraft, Helo, Vertical takeoff aircraft, Air ambulance, Military helicopter, Rescue helicopter |
| Small Car | Compact car, Sedan, Hatchback, Subcompact vehicle, Economy car, City car, Two-door car, Four-door car |
| Bus | Coach, Public transport bus, City bus, School bus, Shuttle bus, Tour bus, Transit vehicle |
| Pickup Truck | Pickup, Light-duty truck, Flatbed pickup, Open-bed truck, Utility pickup, Crew cab truck |
| Truck w/Box | Box van, Enclosed cargo truck, Panel truck, Cube van, Moving truck, Dry van |
| Truck w/Flatbed | Flatbed truck, Open-deck truck, Platform truck, Stake bed truck, Flatbed lorry |
| Truck w/Liquid | Tanker truck, Fuel truck, Liquid transport truck, Water tanker, Milk truck, Chemical tanker |
| Crane Truck | Mobile crane, Truck-mounted crane, Boom truck, Crane lorry, Hydraulic crane truck |
| Railway Vehicle | Train car, Railcar, Rolling stock, Train unit, Rail vehicle |
| Passenger Car (Rail) | Coach, Passenger carriage, Sleeper car, Commuter car, Rail coach, Transit car |
| Cargo Car (Rail) | Freight car, Goods wagon, Boxcar, Hopper car, Gondola |
| Flat Car (Rail) | Flatbed railcar, Flat wagon, Bulkhead flatcar, Container flatcar, Platform railcar |
| Tank Car (Rail) | Tanker railcar, Liquid cargo car, Chemical tanker, Fuel tanker wagon, Oil tank car |
| Locomotive | Train engine, Rail engine, Diesel locomotive, Electric locomotive, Steam engine, Lead unit |
| Motorboat | Speedboat, Powerboat, Motorized vessel, Runabout, Dinghy, Patrol boat |
| Tugboat | Tug, Harbor tug, Towboat, Pusher tug, Assist vessel |
| Barge | Flat-bottomed boat, Cargo barge, Canal barge, Hopper barge, Pontoon |
| Fishing Vessel | Fishing boat, Trawler, Fishing trawler, Commercial fishing ship, Longliner, Gillnetter |
| Ferry | Passenger ferry, Car ferry, Ro-Ro ferry, River ferry, Shuttle boat |

Table 11: Sensitivity of zero-shot mAP (%) to the number of synonyms per class in the multi-instance text-bag. "full" corresponds to the 4–9 synonyms used in the main paper.

| Dataset | $k = 3$ | $k = $ full | $k = 9$ |
|---|---|---|---|
| DOTAv2 | 38.1 | **38.6** | 38.5 |
| VisDrone (Images) | 44.2 | **44.9** | 44.7 |
| HRRSD | 73.0 | 74.1 | **74.2** |

We report zero-shot mAP on DOTAv2, VisDrone (Images), and HRRSD in Table 11. The variation across configurations is consistently small, indicating that the method is not sensitive to the exact number of synonyms per class.

## G  CATASTROPHIC FORGETTING ON FULL COCO

To more rigorously assess catastrophic forgetting on ground-view images, we evaluate all methods on the full set of COCO categories. As shown in Table 12, naively fine-tuning OWLv2 on aerial images causes substantial forgetting of ground-view semantics: the zero-shot mAP on COCO drops from 56.0 to 32.8 (a decrease of 23.2 mAP).

In contrast, our cross-view alignment strategy preserves the frozen ground-view encoder and largely retains ground-level performance, achieving 53.1 mAP on COCO. This recovers 20.3 mAP of the

Table 12: Catastrophic forgetting evaluation on ground-view images using all COCO categories. Fine-tuning OWLv2 on aerial-only data leads to a large mAP drop on COCO, while our cross-view alignment retains most of the original performance.

| Method | mAP |
|---|---|
| OWLv2 zero-shot (no training) | 56.0 |
| OWLv2 fine-tuned on aerial only | |
|     (catastrophic forgetting: $-23.2$) | 32.8 |
| Ours: OWLv2 + cross-view alignment | |
|     (recovers ground-view semantics: $+20.3$) | 53.1 |

lost performance relative to the fine-tuned model, demonstrating that our method substantially mitigates catastrophic forgetting while still adapting to the aerial domain.

## H    COMPARISON WITH REMOTECLIP

We additionally compare our method to RemoteCLIP, a CLIP variant further fine-tuned on large-scale aerial imagery (xView). As expected, RemoteCLIP achieves strong performance on xView, the dataset it is directly fine-tuned on. However, when evaluated zero-shot on other aerial benchmarks (DOTAv2, VisDrone, DIOR, HRRSD), its performance drops substantially.

In contrast, our cross-view alignment approach, which does not use any labels from these target benchmarks, maintains strong zero-shot performance across datasets, indicating better cross-dataset generalization and making it a more robust open-vocabulary solution. The results are summarized in Table 13.

Table 13: Cross-dataset generalization of RemoteCLIP vs. our method. RemoteCLIP is fine-tuned on xView, where it performs well, but generalizes poorly to other aerial benchmarks. Our method maintains strong zero-shot performance.

| Dataset (test) | RemoteCLIP | Ours |
|---|---|---|
| *Finetuned* | | |
| xView | 48.00 | 37.91 |
| *Zero-Shot* | | |
| DOTAv2 | 18.00 | 38.60 |
| VisDrone (Images) | 12.40 | 44.97 |
| VisDrone (Videos) | 10.20 | 37.02 |
| DIOR | 16.80 | 63.91 |
| HRRSD | 20.30 | 74.12 |

## I    COMPARISON WITH LATEST OPEN-VOCABULARY DETECTORS

To address concerns about outdated baselines, we additionally compare our approach with several recent state-of-the-art open-vocabulary detectors on the VisDrone (Images) benchmark. As shown in Table 14, our method achieves the best performance, surpassing GLIPv2, GroundingDINO 1.5, and YOLO-World V2.1 in terms of mAP.

Table 14: State-of-the-art comparison on VisDrone (Images) against recent open-vocabulary detection methods.

| Model | mAP |
|---|---|
| GLIPv2 | 36.8 |
| GroundingDINO 1.5 | 40.1 |
| YOLO-World V2.1 | 42.7 |
| Ours | 44.9 |

