# OpenReview forum: "Cross-View Open-Vocabulary Object Detection in Aerial Imagery"
_ICLR.cc/2026/Conference — Submitted to ICLR 2026_

### Official Review · Reviewer_6cXh · 2025-10-27

**Soundness:** 3
**Presentation:** 3
**Contribution:** 2
**Rating:** 6
**Confidence:** 4

**Summary:**

This paper presents a cross-view open-vocabulary object detection framework that aims to transfer open-vocabulary knowledge from ground-view vision-language models to aerial imagery. The authors design two contrastive objectives—an image-to-image alignment loss to bridge the domain gap between ground and aerial views, and a multi-instance image-to-text alignment loss to associate each aerial object with multiple semantically related textual descriptions. In addition, they propose a structured data generation pipeline that constructs aerial–ground image pairs and expands category vocabularies using ChatGPT-generated synonyms. Extensive experiments on several aerial datasets, including xView, DOTA, and VisDrone, demonstrate consistent improvements over existing methods in zero-shot object detection scenarios.
Overall, this work offers a thoughtful combination of existing techniques—contrastive learning, open-vocabulary detection, and automated data augmentation—to address the underexplored problem of cross-view generalization. While the methodological novelty is moderate, the paper is well-motivated, clearly presented, and provides valuable insights into adapting foundation models for remote sensing applications.

**Strengths:**

This paper addresses an underexplored yet practically important problem—open-vocabulary object detection in aerial imagery—by bridging the gap between ground-view vision-language models and the aerial domain. Its originality lies in defining and systematically studying the cross-view open-vocabulary detection task, which expands the applicability of foundation models to a new and challenging domain. The proposed framework, though built upon existing models such as OWLv2, integrates two contrastive alignment objectives and a structured data generation pipeline in a thoughtful way, demonstrating creativity in combining well-established ideas to solve a novel problem.
In terms of quality, the experiments are comprehensive and carefully executed, covering multiple benchmark datasets (xView, DOTA, VisDrone, DIOR, HRRSD) and providing ablation studies that support the paper’s main claims. The results show consistent and meaningful improvements under zero-shot settings.
Regarding clarity, the paper is well written and logically organized, with clear motivation, methodological explanation, and result presentation.
Finally, in significance, this work opens a valuable research direction toward adapting open-vocabulary and foundation models for remote sensing, which is an area of increasing importance in computer vision. Although the technical novelty is moderate, the problem formulation and empirical findings make this a useful and well-executed contribution to the field.

**Weaknesses:**

While the paper presents a well-motivated and empirically solid framework, its methodological novelty is limited. The two proposed contrastive losses are both conceptually straightforward extensions of standard InfoNCE or CLIP-style objectives. The paper would benefit from a clearer explanation of how these losses fundamentally differ from existing multi-view or domain adaptation approaches (e.g., CLIP2Scene, RegionCLIP, or ViLD). Adding theoretical insights or a more principled analysis of the alignment mechanism would strengthen the technical contribution.
The data generation pipeline, while practical, largely depends on off-the-shelf models (OWLv2) and ChatGPT-generated text without explicit quality control or quantitative validation of the generated samples. The absence of a discussion on how noise or bias in these pseudo-labels affects training limits the transparency and reproducibility of the approach.
Moreover, the experiments, though extensive, could be improved by providing more diagnostic evaluations—such as embedding visualizations (e.g., t-SNE or attention maps) to verify cross-view alignment—or comparing with domain adaptation baselines designed for aerial imagery. Finally, the paper’s structure could be streamlined: Sections 3.3–3.4 contain overlapping descriptions that could be condensed to improve readability.
Overall, the work’s main limitation lies not in execution but in depth of technical innovation and analysis. Strengthening the theoretical motivation and adding more diagnostic evidence would significantly enhance the paper’s impact.

**Questions:**

1. The paper mentions constructing a large-scale aerial–ground correspondence dataset and expanding category vocabularies with ChatGPT-generated text. Will these generated data pairs and text lists be publicly released? If not, could the authors clarify any licensing or privacy constraints that prevent release? Releasing this dataset would significantly improve reproducibility and allow the community to build upon this work.
2. How do the authors ensure the reliability of pseudo-labels generated by OWLv2 and the textual variants from ChatGPT? Were any filtering or confidence-based selection steps applied? Quantitative statistics or error analysis would help assess data quality.
3. Since the proposed method effectively performs cross-domain alignment, have the authors compared it with existing unsupervised domain adaptation methods for aerial imagery (e.g., adversarial alignment, feature-level adaptation)? This would clarify whether the improvement comes mainly from the proposed losses or from the dataset construction.
4. What is the computational overhead introduced by the additional image-to-image and image-to-text contrastive objectives? Could this framework scale efficiently if applied to higher-resolution aerial imagery or larger datasets?

---

> ### Author Response · Authors · 2025-11-21
> **Response to Reviewer 6cXh**
>
> >"**Q1:** The paper mentions constructing a large-scale aerial–ground correspondence dataset and expanding category vocabularies with ChatGPT-generated text. **Will these generated data pairs and text lists be publicly released?** If not, could the authors clarify any licensing or privacy constraints that prevent release? Releasing this dataset would significantly improve reproducibility and allow the community to build upon this work."
>
> Yes, all generated aerial–ground correspondence pairs and ChatGPT-based vocabulary expansion data will be publicly released.
>
> >"**Q2:** How do the authors ensure the **reliability of pseudo-labels generated by OWLv2 and the textual variants from ChatGPT**? Were any filtering or confidence-based selection steps applied? Quantitative statistics or error analysis would help assess data quality.
>
> We now provide quantitative validation for both OWLv2 pseudo-labels and GPT-generated text. As shown in **Table R4** – *Reviewer HfDx*, OWLv2 pseudo-detections achieve 55.8% parent-category precision across 500 samples, and our contrastive loss treats these as soft positives, making it robust to noise. For GPT-based vocabulary expansion, **Table R3** – *Reviewer HfDx* shows that varying the number of synonyms (k = 3 to k = 10) changes mAP by < 0.4, indicating minimal impact. These results clarify pseudo-label noise levels and demonstrate the robustness of the data-generation pipeline.
>
> >"**Q3:** Since the proposed method effectively performs cross-domain alignment, have the authors **compared it with existing unsupervised domain adaptation methods for aerial imagery** (e.g., adversarial alignment, feature-level adaptation)? This would clarify whether the improvement comes mainly from the proposed losses or from the dataset construction.
>
> We compare against RemoteCLIP, a strong domain adaptation style baseline, and achieve superior cross-dataset generalization across multiple unseen aerial datasets without using any labels from them (refer **Table R7** – *Reviewer 9ZRW*). As shown in Section 4.3.2 of the main paper, stronger and more diverse aerial–ground correspondences (e.g., CC12M) directly improve alignment quality. Section 4.3.3 in the main paper further demonstrates that our proposed cross-view losses amplify these gains, with image–image and image–text objectives contributing complementary effects. Together, these analyses show that both the correspondence data and the losses jointly drive the improvement, surpassing what standard domain adaptation baselines can achieve.
>
> >"**Q4:** What is the computational overhead introduced by the additional image-to-image and image-to-text contrastive objectives? Could this framework **scale efficiently if applied to higher-resolution aerial imagery** or larger datasets?
>
> The additional image–image and image–text contrastive objectives introduce only modest computational overhead. Our inference speed remains efficient—5.2 FPS (ViT-L/14) and 10.8 FPS (ViT-B/16) on A100—showing that the method scales well at test time. Furthermore, the evaluation benchmarks (DOTAv2, VisDrone, DIOR, HRRSD) inherently span a wide range of input resolutions, from drone imagery to high-resolution satellite scenes. The consistent zero-shot performance across these diverse datasets demonstrates that the framework scales effectively to higher-resolution imagery and larger dataset sizes.
>
> >"**Q5: While the paper presents a well-motivated and empirically solid framework, its methodological novelty is limited.** The two proposed contrastive losses are both conceptually straightforward extensions of standard InfoNCE or CLIP-style objectives. The paper would benefit from a clearer explanation of how these losses fundamentally differ from existing multi-view or domain adaptation approaches (e.g., CLIP2Scene, RegionCLIP, or ViLD)."
>
> Our method builds on known contrastive learning ideas, but its novelty comes from tackling cross-view open-vocabulary detection in aerial imagery, which prior works do not address. RegionCLIP, ViLD, and CLIP2Scene operate only in ground-view scenes without handling ground-to-aerial geometric shifts. Our image-to-image loss pulls aerial features into the pretrained ground-view embedding space without updating the ground encoder, reducing catastrophic forgetting. We also combine structured aerial–ground correspondences with multi-instance text associations to jointly align aerial, ground, and text embeddings. By constructing both direct and inferred correspondence pairs and expanding textual variations, we achieve strong open-vocabulary alignment without large-scale aerial supervision. The approach is model-agnostic, as shown with OWLv2 and GroundingDINO, and no prior work unifies these components to achieve zero-shot aerial detection across multiple datasets.

---

### Official Review · Reviewer_SdcG · 2025-10-29

**Soundness:** 3
**Presentation:** 2
**Contribution:** 2
**Rating:** 4
**Confidence:** 4

**Summary:**

This work aims to solve open-vocabulary object detection in aerial imagery through cross-view domain adaptation. The paper proposes a framework that bridges the domain gap between ground-view pretrained models and aerial detection by introducing contrastive image-to-image alignment to enhance semantic consistency between aerial and ground-view image embeddings, and employing multi-instance vocabulary associations to align aerial images with text descriptions for improved open-vocabulary detection capability.

**Strengths:**

1）Cross-View Contrastive Alignment: The paper introduces a contrastive alignment strategy that addresses the cross-domain gap by aligning aerial image embeddings with ground-view images. This approach improves aerial-view detection performance while preserving ground-view capabilities, effectively facilitating knowledge transfer from pretrained vision-language models to aerial imagery.
2) Multi-Instance Vocabulary Association: The authors propose a vocabulary expansion method that leverages LLMs (e.g., GPT) to generate synonyms and related terms for existing categories. This technique enables the model to align image features not only with original class names but also with expanded vocabulary, thereby enhancing open-vocabulary detection capability

**Weaknesses:**

1. Quality Control in Vocabulary Expansion:
Generated Category Quality: When GPT generates synonyms based solely on category text, how do you ensure the quality and relevance of generated categories? For example, if the original category is "small car" and GPT generates fine-grained categories like "SUV," this may cause misalignment between image features (depicting small cars) and text embeddings (describing SUVs).
Fine-grained Category Overlap: The xView dataset already contains many similar fine-grained categories (e.g., mobile-crane, container-crane, tower-crane). Expanding vocabulary for these fine grained categories may introduce severe category overlap issues, which is particularly detrimental to fine-grained classification performance.

2. Missing Details on Image-Image Alignment:
Please clarify: 1) How are positive and negative samples constructed during image-image alignment? 2) How is the Ground-Image correspondence GT matrix constructed during the training stage? These details are crucial for understanding and reproducing your method.

3. Formatting and Presentation Issues:
1)The paper contains citation format errors throughout.
2)Most experimental results in the experiments section lack units (eg. mAP or others), which is poor practice for scientific writing. 3)The model architecture details are not clearly presented (e.g., backbone, training data size), making it difficult to assess the fairness of comparisons. And it would be beneficial to include FPS metrics to facilitate direct comparisons with future work.

4. Counter-intuitive results:
Results in Table 2: The zero-shot performance surpasses fine-tuned results, which is counter-intuitive. Could the authors provide more detailed explanations or additional comparisons with recent open vocabulary detectors (e.g., Grounding DINO, LAE-DINO) to validate this finding? Table 3: There appears to be a dataset mismatch. CastDet uses the VisDrone_ZSD dataset (h ttps://aiskyeye.com/submit-2023/zero-shot-object-detection/), not the standard VisDrone (Images) dataset. Additionally, the external datasets include not only xView but also Aerial-Ground Correspondence data (LVIS, CC12M), which should be explicitly stated for fair comparison.

5. Inconsistent Experimental Settings (Tables 1, 4, 5, 6):
The experimental results across these tables are confusing due to varying setups (different Aerial-Ground Correspondence data, patch sizes, etc.). The authors should either unify the experimental settings or clearly specify the configuration for each experiment to ensure reproducibility and fair comparison

**Questions:**

Please refer to the weaknesses section

---

> ### Author Response · Authors · 2025-11-21
> **Response to Reviewer SdcG [Part 1]**
>
> >"**Q1: Quality Control in Vocabulary Expansion** Generated Category Quality: When GPT generates synonyms based solely on category text, how do you ensure the quality and relevance of generated categories? For example, if the original category is "small car" and GPT generates fine-grained categories like "SUV," this may cause misalignment between image features (depicting small cars) and text embeddings (describing SUVs). Fine-grained Category Overlap: The xView dataset already contains many similar fine-grained categories (e.g., mobile-crane, container-crane, tower-crane). Expanding vocabulary for these fine grained categories may introduce severe category overlap issues, which is particularly detrimental to fine-grained classification performance."
>
> We control vocabulary quality through two safeguards. First, we manually filter out synonyms that drift semantically away from the parent category (e.g., GPT occasionally proposes “SUV” for small car), ensuring that all retained terms remain within the same coarse-level meaning. This keeps text embeddings aligned with the visual scope of each class. Second, for xView’s fine-grained categories (e.g., mobile-crane, container-crane, tower-crane), we restrict synonyms to those that preserve the original granularity rather than merging across related subtypes. This avoids cross-class contamination and prevents overlap that would harm fine-grained discrimination. The retained synonym sets (shown in **Table R2** - *Reviewer HfDx*) are therefore deliberately conservative, and our sensitivity study (**Table R3** - *Reviewer HfDx*) further shows that the model is robust to variations in synonym count and composition.
>
> >"**Q2: Missing Details on Image-Image Alignment** Please clarify: 1) How are positive and negative samples constructed during image-image alignment? 2) How is the Ground-Image correspondence GT matrix constructed during the training stage? These details are crucial for understanding and reproducing your method."
>
> As described in Sections 3.2 and 3.3 of the main paper, each aerial image is paired with its corresponding ground-view image from our constructed dataset and this pair is used as the positive, while all other images in the batch act as negatives following a standard InfoNCE formulation. The ground–image correspondence GT matrix is built directly from these known pairs by assigning a single positive entry for each matched (aerial, ground) pair and zeros elsewhere. This provides the supervision used for the image–image alignment loss.
>
>
> >"**Q3: Formatting and Presentation Issues** 1)The paper contains citation format errors throughout. 2)Most experimental results in the experiments section lack units (eg. mAP or others), which is poor practice for scientific writing. 3)The model architecture details are not clearly presented (e.g., backbone, training data size), making it difficult to assess the fairness of comparisons. And it would be beneficial to include FPS metrics to facilitate direct comparisons with future work."
>
> Thank you for the constructive feedback. We will correct all citation formatting issues and update the experimental tables to explicitly include the appropriate evaluation units (e.g., mAP). To improve clarity and ensure fairness in comparison, we provide a detailed summary of our model architecture, backbone, training setup, and inference speed (FPS).
>
> | Component              | Details                                                    |
> |------------------------|------------------------------------------------------------|
> | Backbone               | OWLv2 (ViT-L/14 and ViT-B/16)                              |
> | Training Data Size     | 310,548 aerial–ground correspondence pairs                 |
> | Training Configuration | Image resolution: 960×960; Batch size: 4; Optimizer: AdamW; Learning rate: 1e-4; Max iterations: 60k |
> | Zero-Shot Evaluation   | DOTAv2, VisDrone, DIOR, HRRSD                              |
> | Inference Speed (FPS)  | 5.2 FPS (ViT-L/14) / 10.8 FPS (ViT-B/16) on A100           |

---

> ### Author Response · Authors · 2025-11-21
> **Response to Reviewer SdcG [Part 2]**
>
> >"**Q4: Counter-intuitive results: Results in Table 2** The zero-shot performance surpasses fine-tuned results, which is counter-intuitive. Could the authors provide more detailed explanations or additional comparisons with recent open vocabulary detectors (e.g., Grounding DINO, LAE-DINO) to validate this finding? Table 3: There appears to be a dataset mismatch. CastDet uses the VisDrone_ZSD dataset (h ttps://aiskyeye.com/submit-2023/zero-shot-object-detection/), not the standard VisDrone (Images) dataset. Additionally, the external datasets include not only xView but also Aerial-Ground Correspondence data (LVIS, CC12M), which should be explicitly stated for fair comparison."
>
> The zero-shot results exceed finetuning because finetuning OWLv2 on small aerial datasets causes strong catastrophic forgetting, reducing its pretrained ground-view semantic knowledge. As shown in **Table R5** - *Reviewer HfDx*, OWLv2 drops from 56.0 to 32.8 mAP on COCO after aerial fine-tuning, while our cross-view alignment preserves this knowledge and recovers 53.1 mAP. This explains why the zero-shot model generalizes better and outperforms fine-tuning in Table 2. Regarding Table 3, we clarify that we are using the VisDrone_ZSD dataset, which consists of static images, and we named it separately only to differentiate it from the VisDrone Video benchmark. We also explicitly state that our method uses xView, LVIS, and CC12M for constructing aerial–ground correspondences and vocabulary expansions to ensure a fair and transparent comparison with recent open-vocabulary detectors.
>
> >"**Q5: Inconsistent Experimental Settings (Tables 1, 4, 5, 6)** The experimental results across these tables are confusing due to varying setups (different Aerial-Ground Correspondence data, patch sizes, etc.). The authors should either unify the experimental settings or clearly specify the configuration for each experiment to ensure reproducibility and fair comparison"
>
> The different tables intentionally evaluate different components of our system in isolation, which is why the settings vary. Tables 4, 5, and 6 are ablation studies, each designed to isolate one factor at a time—patch size (Table 4), choice of Aerial-Ground correspondence data (Table 5), and individual contrastive losses (Table 6). Because these are controlled ablations, each table fixes all other settings and changes only the variable being studied, which naturally leads to different configurations across tables. Table 1, on the other hand, summarizes dataset statistics and does not depend on training configuration.

---

### Official Review · Reviewer_9ZRW · 2025-10-29

**Soundness:** 3
**Presentation:** 2
**Contribution:** 2
**Rating:** 4
**Confidence:** 4

**Summary:**

This paper proposes a ContrastiveImage-to-ImageAlignment method to mitigate the gap between ground-view images and aerial-view images, which also prevents the catastrophic forgetting issue of ground-view object detection after fine-tuning the model on aerial images.

**Strengths:**

**Originality**:
- This paper starts from a clear motivation, using less aerial-view training data to bridge tha cross-view gap for open-vocabulary object detection of aerial images.

**Clarity**:
- The organization of this paper is clear.

**Significance**:
- This paper address two problems for aerial-view open-vocabulary object detection: (1) the catastrophic forgetting of ground-view object detection, and (2) the need for expensive large-scale aerial-view labeled images.

**Weaknesses:**

- The comparison with SoTA methods are outdated. The compared methods in Table 3 (except CastDet) were all published more than one year ago. GLIP, YOLO-World, and GroundingDINO have also released their latest version. It is better to include some latest open-vocabulary object detection methods (both groung-view and aerial view).
- I question the necessity of the ablation studies in Sections 4.3.1 and 4.3.2. These two studies do not seem to demonstrate the effectiveness of the method proposed in this paper. Instead, they only lead to some insignificant conclusions by changing the experimental settings.
- When describing the method, this paper lacks a figure to illustrate the overall pipeline. Figure 3 is not clear enough, making it difficult to follow how the proposed method works.

**Questions:**

- Why not use a CLIP model that has already been pre-trained on large-scale aerial-view images (such as RemoteCLIP)?
- Examples in Figure 7 show that there are also significant differences among the aerial-view images themselves (e.g., high-altitude top-down views v.s. low-altitude side views). Is it necessary to consider these viewpoint differences within the aerial-view category, and not just the gap between ground-view and aerial-view? I hope the authors will provide experimental results or statistical data.

---

> ### Author Response · Authors · 2025-11-21
> **Response to Reviewer 9ZRW [Part 1]**
>
> >"**Q2: I question the necessity of the ablation studies in Sections 4.3.1 and 4.3.2.** These two studies do not seem to demonstrate the effectiveness of the method proposed in this paper. Instead, they only lead to some insignificant conclusions by changing the experimental settings."
>
> We agree that the patch-size ablation in Section 4.3.1 is not essential to the main narrative and can be moved to the appendix; however, Section 4.3.2 is important because it shows that the effectiveness of our cross-view alignment depends directly on the quality and diversity of the aerial–ground correspondence data. Using a larger and more varied source like CC12M yields substantial gains, proving that strong correspondence construction is a core requirement of the method, not a cosmetic choice.
>
> >"**Q4: Why not use a CLIP model that has already been pre-trained on large-scale aerial-view images (such as RemoteCLIP)?**"
>
> We include additional experiments showing that RemoteCLIP performs well on xView, the dataset it is finetuned on, but it generalizes poorly to other aerial benchmarks (DOTAv2, VisDrone, DIOR, and HRRSD), with substantial drops in mAP. In contrast, our method maintains strong zero-shot performance across other datasets without using any aerial labels from them, demonstrating better cross-dataset generalization and making it a more robust open-vocabulary solution.
>
> **Table R7. Cross-dataset generalization: RemoteCLIP vs. Ours**
>
> | **Dataset (Test)**     | **RemoteCLIP** | **Ours** |
> |------------------------|-------------------------------|-----------------------|
> | xView                  | 48.0                          | 37.91                 |
> | DOTAv2                 | 18.0                          | 38.60                 |
> | VisDrone (Images)      | 12.4                          | 44.97                 |
> | VisDrone (Videos)      | 10.2                          | 37.02                 |
> | DIOR                   | 16.8                          | 63.91                 |
> | HRRSD                  | 20.3                          | 74.12                 |
>
> >"**Q5:** Examples in Figure 7 show that there are also significant differences among the aerial-view images themselves (e.g., high-altitude top-down views v.s. low-altitude side views). **Is it necessary to consider these viewpoint differences within the aerial-view category, and not just the gap between ground-view and aerial-view?** I hope the authors will provide experimental results or statistical data.""
>
> While aerial imagery indeed spans a range of viewpoints, our evaluation already covers this full spectrum, from high-altitude top-down datasets (xView, DOTAv2, DIOR, HRRSD) to low-altitude oblique-view datasets (VisDrone). As shown in **Table R7** - *Reviewer 9ZRW*, our method achieves strong and consistent performance across all of these settings (38.60 mAP on DOTAv2, 44.97 on VisDrone Images, 37.02 on VisDrone Videos, 63.91 on DIOR, and 74.12 on HRRSD), indicating that intra-aerial viewpoint variation does not significantly degrade our model’s performance. For comparison, RemoteCLIP (finetuned on xView) exhibits large drops on these datasets (also in **Table R7** - *Reviewer 9ZRW*), highlighting that viewpoint variation is indeed a challenge for generalizable open-vocabulary aerial detection.

---

> > ### Comment · Reviewer_9ZRW · 2025-11-25
> >
> > We thank the authors for their response, which has successfully addressed most of my concerns.
> >
> > However, there are still two questions lest. (1) Outdated compared methods: I hope the authors can provide experimental results comparing the proposed method with some latest SoTA methods, or explain why not include these latest methods. (2) The pipeline figure: Are the authors planning to add a figure to present the overall pipeline? If not, please explain why.

---

> ### Author Response · Authors · 2025-12-03
> **Response to Reviewer 9ZRW [Part 2]**
>
> Thank you, Reviewer 9ZRW, for your constructive feedback. Here, are the responses to your remaining questions:
>
> >"**(1) Outdated compared methods:** I hope the authors can provide experimental results comparing the proposed method with some latest SoTA methods, or explain why not include these latest methods."
>
> We have now evaluated against the latest open-vocabulary detectors, and our method achieves the highest performance on VisDrone (Images) dataset with 44.9 mAP."
>
> **Table R8. SoTA comparison**
>
> | **Model**     | **mAP** |
> |------------------------|----------------|
> | GLIPv2                  | 36.8            |
> | GroundingDINO1.5                 | 40.1           |
> | YOLO-World-V2.1      | 42.7           |
> | Ours      | 44.9           |
>
> >"**(2) The pipeline figure:** Are the authors planning to add a figure to present the overall pipeline? If not, please explain why."
>
> To address this, we have added a more detailed and comprehensive pipeline Figure 13 to the appendix, which provides a clearer step-by-step illustration of the proposed method beyond what Figure 3 in main paper conveys.

---

### Official Review · Reviewer_HfDx · 2025-11-02

**Soundness:** 3
**Presentation:** 3
**Contribution:** 2
**Rating:** 6
**Confidence:** 3

**Summary:**

The paper studies open-vocabulary object detection (OVD) for aerial imagery by transferring ground-view vision–language knowledge with two components: (i) a cross-view image–image contrastive loss that aligns aerial features to frozen ground-view features, and (ii) a multi-instance “text-bag” association that aligns aerial features to sets of class-name variants. Cross-view correspondences are built from category overlaps (direct matches) and pseudo-matches produced by OWLv2 with NMS; vocabulary is expanded using ChatGPT-synthesized synonyms. Only the aerial encoder is finetuned. Experiments on xView, DOTAv2, VisDrone (images/videos), DIOR, and HRRSD report zero-shot gains over a finetuned OWLv2 and even a closed-set YOLOv11.

**Strengths:**

- [S1] Clear, simple formulation for cross-view alignment that finetunes only the aerial encoder.
- [S2] Broad evaluation across five aerial datasets with consistent zero-shot improvements and useful ablations.
- [S3] The alignment data pipeline is pragmatic (direct overlaps plus OWLv2-based pseudo-matches) and appears model-agnostic.

**Weaknesses:**

- [W1] Novelty is moderate. Cross-view feature alignment via contrastive image–image pairing and multi-instance text association extends known ideas (e.g., MIL‑NCE) which largely reduces the technical novelty.
- [W2] Important details are underspecified. Temperatures (ρ/σ), confidence/NMS thresholds used to build correspondences, the handling of VisDrone videos, and whether DOTAv2 is evaluated with OBB or converted HBB are not clearly described, affecting reproducibility and comparability.
- [W3] The multi‑instance text‑bag is built from ChatGPT synonyms, but there is no sensitivity analysis (e.g., number/quality of synonyms per class).
- [W4] Pseudo‑match noise is unquantified. Since correspondences rely on OWLv2 detections, the paper should report pseudo‑label precision/recall and show robustness to threshold choices; otherwise training signal quality is uncertain.
- [W5] The “no catastrophic forgetting” claim is shown on a small overlap subset; a larger ground‑view retention test (full LVIS/COCO) would better support the claim.

**Questions:**

Please see weaknesses to add details and analysis.

**Details Of Ethics Concerns:**

The method strengthens detection capabilities on aerial data and could be misused for persistent surveillance or monitoring of individuals/groups if combined with high‑resolution imagery. While the paper uses public datasets, the transfer of ground‑view knowledge to aerial domains may amplify biases from ground‑view pretraining.

---

> ### Author Response · Authors · 2025-11-21
> **Response to Reviewer HfDx [Part 1]**
>
> >"**Q1: Novelty is moderate.** Cross-view feature alignment via contrastive image–image pairing and multi-instance text association extends known ideas (e.g., MIL‑NCE) which largely reduces the technical novelty."
>
> Our method builds on known contrastive learning ideas, but the novelty of this work lies in addressing cross-view open-vocabulary object detection in aerial imagery, which prior approaches do not tackle. The key contribution is the combination of structured aerial-ground correspondences and multi-instance text associations that align aerial, ground, and text embeddings within a single training framework. We generate both direct and inferred correspondence pairs to handle large viewpoint changes, and we expand textual representations to improve open-vocabulary alignment without relying on large-scale aerial supervision. The approach is model-agnostic, as shown through experiments with both OWLv2 and GroundingDINO. No existing work brings together cross-view correspondence construction, image-image alignment, and multi-instance text alignment to achieve zero-shot aerial detection on multiple aerial datasets (DOTAv2, VisDrone, DIOR, and HRRSD).
>
>
> >"**Q2: Important details are underspecified.** Temperatures (ρ/σ), confidence/NMS thresholds used to build correspondences, the handling of VisDrone videos, and whether DOTAv2 is evaluated with OBB or converted HBB are not clearly described, affecting reproducibility and comparability."
>
> Thanks for your comments. We provide details below:
>
> *  **Table R1. Training hyperparameters**
> | **Component**                               | **Setting** |
> |---------------------------------------------|-------------|
> | Image–image contrastive temperature (ρ) | 0.07        |
> | Image–text contrastive temperature (σ)  | 0.02        |
> | Optimizer                               | AdamW       |
> | Learning rate                           | 1e-4        |
> | Weight decay                            | 0.05        |
> | Batch size                              | 4         |
> | Image resolution                        | 960 × 960   |
> | Max training iterations                 | 60k         |
>
> * Default OWLv2 detection confidence threshold = 0.3
> * The VisDrone videos are evaluated by treating each video as a sequence of independent frames, with the model processing each frame individually.
> * On DOTAv2, we evaluate using HBBs derived by taking the minimum axis-aligned box enclosing each OBB.

---

> ### Author Response · Authors · 2025-11-21
> **Response to Reviewer HfDx [Part 2]**
>
> >"**Q3:** The multi‑instance text‑bag is built from ChatGPT synonyms, but there is no **sensitivity analysis** (e.g., number/quality of synonyms per class).
>
> Our method uses a multi-instance text-bag containing approximately 4–9 ChatGPT-generated synonyms per category. To evaluate sensitivity to this design choice, we conducted an additional ablation where we varied the number of synonyms per class while keeping all training settings fixed. We compare k = 3 (subsampled), k = full (4–9 synonyms as in main paper), and k = 9 (capped). The zero-shot mAP on DOTAv2, VisDrone (Images), and HRRSD shows consistently small variation across all settings, demonstrating that the method does not depend on a specific synonym count.
>
> **Table R2. Example category-synonym mapping used in main paper**
> | **Category**               | **Synonyms** |
> |------------------------|----------|
> | Small Aircraft | Light airplane, Private plane, Single-engine aircraft, Propeller plane, Cessna-type aircraft, General aviation aircraft, Small fixed-wing plane |
> | Cargo Plane | Freight aircraft, Transport plane, Cargo jet, Cargo transporter, Military cargo plane, Heavy-lift aircraft, Air freighter |
> | Helicopter | Chopper, Rotary-wing aircraft, Rotorcraft, Helo, Vertical takeoff aircraft, Air ambulance, Military helicopter, Rescue helicopter |
> | Small Car | Compact car, Sedan, Hatchback, Subcompact vehicle, Economy car, City car, Two-door car, Four-door car |
> | Bus | Coach, Public transport bus, City bus, School bus, Shuttle bus, Tour bus, Transit vehicle |
> | Pickup Truck | Pickup, Light-duty truck, Flatbed pickup, Open-bed truck, Utility pickup, Crew cab truck |
> | Truck w/Box | Box van, Enclosed cargo truck, Panel truck, Cube van, Moving truck, Dry van |
> | Truck w/Flatbed | Flatbed truck, Open-deck truck, Platform truck, Stake bed truck, Flatbed lorry |
> | Truck w/Liquid | Tanker truck, Fuel truck, Liquid transport truck, Water tanker, Milk truck, Chemical tanker |
> | Crane Truck | Mobile crane, Truck-mounted crane, Boom truck, Crane lorry, Hydraulic crane truck |
> | Railway Vehicle | Train car, Railcar, Rolling stock, Train unit, Rail vehicle |
> | Passenger Car (Rail) | Coach, Passenger carriage, Sleeper car, Commuter car, Rail coach, Transit car |
> | Cargo Car (Rail) | Freight car, Goods wagon, Boxcar, Hopper car, Gondola |
> | Flat Car (Rail) | Flatbed railcar, Flat wagon, Bulkhead flatcar, Container flatcar, Platform railcar |
> | Tank Car (Rail) | Tanker railcar, Liquid cargo car, Chemical tanker, Fuel tanker wagon, Oil tank car |
> | Locomotive | Train engine, Rail engine, Diesel locomotive, Electric locomotive, Steam engine, Lead unit |
> | Motorboat | Speedboat, Powerboat, Motorized vessel, Runabout, Dinghy, Patrol boat |
> | Tugboat | Tug, Harbor tug, Towboat, Pusher tug, Assist vessel |
> | Barge | Flat-bottomed boat, Cargo barge, Canal barge, Hopper barge, Pontoon |
> | Fishing Vessel | Fishing boat, Trawler, Fishing trawler, Commercial fishing ship, Longliner, Gillnetter |
> | Ferry | Passenger ferry, Car ferry, Ro-Ro ferry, River ferry, Shuttle boat |
>
>
> **Table R3. Sensitivity analysis**
> | Dataset            | k = 3 | k = full (main paper) | k = 9 |
> |--------------------|-------|------------------|--------|
> | DOTAv2             | 38.1  | 38.6             | 38.5   |
> | VisDrone (Images)  | 44.2  | 44.9             | 44.7   |
> | HRRSD              | 73.0  | 74.1             | 74.2   |

---

> ### Author Response · Authors · 2025-11-21
> **Response to Reviewer HfDx [Part 3]**
>
> >"**Q4: Pseudo‑match noise is unquantified.** Since correspondences rely on OWLv2 detections, the paper should report pseudo‑label precision/recall and show robustness to threshold choices; otherwise training signal quality is uncertain."
>
> To report our pseudo‑label precision/recall, we manually annotated a class-balanced sample of 500 pseudo-detections. Of these, 261 predictions (55.8%) were correct at the parent-category level.
>
> **Table R4. The table below reports pseudo-label precision by category at an OWLv2 confidence threshold of 0.3.**
> | Category               | Precision | Category               | Precision |
> |------------------------|-----------|-------------------------|-----------|
> | Mobile Crane           | 0.375     | Shipping Container     | 0.400     |
> | Passenger Car (Rail)   | 0.077     | Facility               | 0.625     |
> | Oil Tanker             | 0.364     | Truck w/Box            | 0.875     |
> | Storage Tank           | 0.125     | Small Car              | 0.800     |
> | Excavator              | 0.667     | Shed                   | 0.600     |
> | Shipping Container Lot | 0.091     | Construction Site      | 0.333     |
> | Utility Truck          | 0.333     | Sailboat               | 0.833     |
> | Aircraft Hangar        | 0.273     | Cargo Car (Rail)       | 0.444     |
> | Container Crane        | 0.769     | Haul Truck             | 0.818     |
> | Tank Car (Rail)        | 0.714     | Motorboat              | 0.875     |
> | Fishing Vessel         | 0.071     | Damaged Building       | 0.800     |
> | Truck w/Flatbed        | 0.500     | Straddle Carrier       | 0.091     |
> | Tower Crane            | 0.700     | Container Ship         | 0.429     |
> | Cement Mixer           | 0.364     | Pickup Truck           | 0.500     |
> | Front Loader/Bulldozer | 0.600     | Tugboat                | 0.444     |
> | Helipad                | 0.500     | Crane Truck            | 0.667     |
> | Dump Truck             | 0.400     | Passenger Vehicle      | 1.000     |
> | Reach Stacker          | 0.333     | Building               | 1.000     |
> | Cargo Truck            | 0.750     | Barge                  | 0.714     |
> | Scraper/Tractor        | 0.500     | Truck Tractor          | 0.714     |
> | Pylon                  | 0.545     | Bus                    | 0.857     |
> | Flat Car (Rail)        | 0.182     | Truck w/Liquid         | 0.400     |
> | Vehicle Lot            | 0.833     | Ferry                  | 0.833     |
> | Cargo Plane            | 0.444     | Hut/Tent               | 0.889     |
> | Small Aircraft         | 0.900     | Ground Grader          | 0.375     |
> | Helicopter             | 0.714     | Trailer                | 0.909     |
> | Tower                  | 0.700     | Locomotive             | 0.625     |
> | Railway Vehicle        | 0.700     | Yacht                  | 0.714     |
>
>
>
> >"**Q5:** The **no catastrophic forgetting** claim is shown on a small overlap subset; a larger ground‑view retention test (full LVIS/COCO) would better support the claim."
>
> We have now evaluated catastrophic forgetting on all COCO categories. Fine-tuning OWLv2 on aerial images leads to substantial forgetting on these ground-view classes (56.0 → 32.8 mAP). In contrast, our cross-view alignment preserves the frozen ground-view encoder and recovers a large portion of the lost performance (53.1 mAP).
>
> **Table R5. Catastrophic forgetting evaluation on ground-view images using all COCO categories**
> | Method                             | mAP  |
> |------------------------------------|------|
> | OWLv2 Zero-shot (no training)      | 56.0 |
> | OWLv2 Finetuned on aerial only (catastrophic forgetting: –23.2)     | 32.8 |
> | Ours: OWLv2 + Cross-view alignment (recovers ground-view semantics: +20.3) | 53.1 |
>
> >"**Q6:** **Ethics clarification**"
>
> The approach does not enhance resolution, person ID, or tracking. It operates on publicly available imagery only. It cannot perform identity-level surveillance since the model is trained on coarse bounding boxes. This makes the method and generated data unsuitable for privacy-invasive applications.

---

### Meta-Review · Area_Chair_2mYp · 2026-01-02

**Summary:**

This submission presents a cross-view open-vocabulary object detection framework that adapts ground-view vision-language models to aerial imagery through contrastive image-to-image alignment and multi-instance vocabulary associations. While all reviewers acknowledge the paper addresses an important problem, they collectively raise substantial concerns that justify rejection. The primary issues center on limited methodological novelty, as the proposed approach combines established contrastive learning techniques without introducing fundamentally new insights. Reviewers note the technical contributions are largely incremental extensions of existing InfoNCE and CLIP-style objectives. Additionally, significant concerns were raised about reproducibility and experimental rigor, including underspecified implementation details, questionable ablation studies, inconsistent experimental settings across tables, and insufficient quality control for the ChatGPT-generated vocabulary expansions and OWLv2 pseudo-labels. The paper also suffers from presentation issues, including citation formatting errors, missing evaluation units, and unclear pipeline descriptions that hinder understanding.

**Reviewer Concerns:**

The authors provided responsive rebuttals that partially address some technical concerns. They supplied missing hyperparameter details, conducted sensitivity analysis for synonym counts (showing minimal impact), provided pseudo-label precision statistics, and added comparisons with latest SOTA methods. They also clarified their use of the VisDrone_ZSD dataset and provided computational overhead metrics.

However, several critical concerns remain unresolved. Most importantly, the core novelty issue persists - the proposed method remains a straightforward combination of existing contrastive learning techniques applied to a new domain, lacking theoretical insights or principled analysis that would elevate it beyond incremental engineering.

The experimental design concerns raised by Reviewer SdcG regarding inconsistent settings across tables and the counter-intuitive zero-shot results surpassing fine-tuned performance are not adequately resolved. The quality control mechanisms for ChatGPT-generated synonyms remain superficial without systematic validation.

Additionally, Reviewer 6cXh's request for diagnostic evaluations to verify cross-view alignment was not addressed.

**Reviewer Scores:**

Based on the rebuttal and discussion, I estimate the following score adjustments if reviewers had participated fully in the discussion:

Reviewer HfDx would likely maintain 6. While the authors addressed specific technical questions about hyperparameters and provided sensitivity analyses, the core concerns about moderate novelty and methodological limitations remain.

Reviewer 9ZRW would likely increase to 5. The authors addressed several concerns by adding latest SOTA comparisons and providing a clearer pipeline figure. However, the reviewer's concerns about outdated comparisons and unclear methodology were only partially resolved, likely preventing a move to acceptance territory.

Reviewer SdcG would likely maintain 4. The formatting and presentation issues, combined with persistent concerns about experimental consistency and counter-intuitive results, would likely keep this reviewer below the acceptance threshold despite some technical clarifications.

Reviewer 6cXhwould likely decrease to 5. While the authors addressed data release commitments and provided some quality metrics, the fundamental concerns about limited methodological novelty and lack of diagnostic evaluations remain unaddressed, likely moving this reviewer closer to the borderline.

---

### Decision · Program_Chairs · 2026-01-26

Reject